# An Integrated Analysis of Mechanistic Insights into Biomolecular Interactions and Molecular Dynamics of Bio-Inspired Cu(II) and Zn(II) Complexes towards DNA/BSA/SARS-CoV-2 3CL^pro^ by Molecular Docking-Based Virtual Screening and FRET Detection

**DOI:** 10.3390/biom12121883

**Published:** 2022-12-15

**Authors:** Karunganathan Sakthikumar, Bienfait Kabuyaya Isamura, Rui Werner Maçedo Krause

**Affiliations:** 1Department of Chemistry, Center for Chemico- and Biomedicinal Research (CCBR), Faculty of Science, Rhodes University, Grahamstown 6140, South Africa; 2Department of Chemistry, The University of Manchester, Manchester M13 9PL, UK; 3Center for Chemico- and Biomedicinal Research (CCBR), Faculty of Science, Rhodes University, Grahamstown 6140, South Africa

**Keywords:** 1,10-phenanthroline, FRET, DFT, docking, biomolecules, CoV-2 3CL^Pro^

## Abstract

Novel constructed bioactive mixed-ligand complexes (**1b**) [Cu^II^(L)_2_(phen)] and (**2b**) [Zn^II^(L)_2_(phen)] {where, L = 2-(4-morpholinobenzylideneamino)phenol), phen = 1,10-phenanthroline} have been structurally analysed by various analytical and spectroscopic techniques, including, magnetic moments, thermogravimetric analysis, and X-ray crystallography. Various analytical and spectral measurements assigned showed that all complexes appear to have an octahedral geometry. Agar gel electrophoresis’s output demonstrated that the Cu(II) complex (**1b**) had efficient deoxyribonucleic cleavage and complex (**2b**) demonstrated the partial cleavage accomplished with an oxidation agent, which generates spreadable OH**^●^** through the Fenton type mechanism. The DNA binding constants observed from viscosity, UV–Vis spectral, fluorometric, and electrochemical titrations were in the following sequence: (**1b**) > (**2b**) > (**HL**), which suggests that the complexes (**1b**–**2b**) might intercalate DNA, a possibility that is supported by the biothermodynamic measurements. In addition, the observed binding constant results of BSA by electronic absorption and fluorometric titrations indicate that complex (**1b**) revealed the best binding efficacy as compared to complex (**2b**) and free ligand. Interestingly, all compounds are found to interact with BSA through a static approach, as further attested by FRET detection. The DFT and molecular docking calculations were also performed to realize the electronic structure, reactivity, and binding capability of all test samples with CT-DNA, BSA, and the SARS-CoV-2 3CL^Pro^, which revealed the binding energies were in a range of −8.1 to −8.9, −7.5 to −10.5 and −6.7–−8.8 kcal/mol, respectively. The higher reactivity of the complexes than the free ligand is supported by the FMO theory. Among all the observed data for antioxidant properties against DPPH^᛫^, ^᛫^OH, O_2_^−•^ and NO^᛫^ free radicals, complex (**1a**) had the best biological efficacy. The antimicrobial and cytotoxic characteristics of all test compounds have been studied by screening against certain selected microorganisms as well as against A549, HepG2, MCF-7, and NHDF cell lines, respectively. The observed findings revealed that the activity enhances coordination as compared to free ligand via Overtone’s and Tweedy’s chelation mechanisms. This is especially encouraging given that in every case, the experimental findings and theoretical detections were in perfect accord.

## 1. Introduction

Organometallic and bioinorganic chemistry currently play an inevitable role and have a major impact on the drug design and development process, which also covers a wide range of applications in science and technology. They have gained significant attention over the past four decades in the construction and development of new metal-based anti-tumour medications through various synthetic roots that provide enhanced selectivity via non-covalent interaction modalities with DNA [1]. Particularly, transition metal complexes are being extensively explored as new anticancer medicines because of their biocompatible qualities and oxidative nature [2]. Today, cancer is the second largest cause of mortality worldwide behind heart disease, with low- to middle-income countries accounting for 70% of cancer fatalities. In addition, it has been a significant global economic cost, estimated at $1.16 trillion per year. There were an estimated 20 million cancer cases (including nonmelanoma skin cancer) around the world in 2022. Of these, 10.326 million cases were in men and 9.674 million were in women. Almost 10.6 million cancer deaths, or nearly one in six deaths, occurred in 2022. Globally, nearly one cancer case out of every 500 people is diagnosed. According to GLOBOCAN 2020 reports and WCRFI (World Cancer Research Fund International), the global cancer mortality rate is additionally increasing in the range of 0.308–0.400 million per year, including the top five deadliest cancers (breast > lung > colorectal > prostate > stomach). According to the World Health Organization report, there will be an estimated 26 million people with cancer by 2035, which will cause more than 14.5 million fatalities. The mortality rate of cancer is expected to exceed that of communicable diseases in the next ten years [3]. Despite the substantial advancements in cancer prevention, treatment, and accessibility during the many decades, the prevalence of insights and their implications for cancer worldwide has not diminished at all due to their ineffectiveness, lack of selectivity, and high level of toxicity due to covalent interactions of the platinum-coordinated anticancer drugs, which also lead to numerous inherent side-effects to human health [4]. On the other hand, multidrug-resistant infections and cellular-based illnesses are rapidly spreading, raising concerns on a worldwide scale. Therefore, it is crucial to find new active medications that can contest new targets. Therefore, our team has been concentrating on developing and screening antitumor medications with a specific target that reacts by a variety of mechanisms in order to replace the platinum-based medications and resolve their limitations [5,6].

Several literature surveys also reported that 1,10-phenanthroline-coordinated transition metal complexes revealed various biological properties, including probes of nucleic acid structures, chemical nucleases, fluorescence probes, electron transfer systems, etc. Despite 1,10-phenanthroline being a weak donor chelating bidentate ligand and more basic as compared to 2,2′-bipyridine, they have efficient stacking interaction and cleaving ability with DNA base pairs and might have strong anticancer properties and control apoptosis. This is because their stiff planar geometry, electron-poor heteroaromatic, π-acidic characteristics, and their conveniently placed N-atoms interfered in denaturing metalloenzymes are employed for coordination [7]. Nowadays, morpholine-linked gefitinib and linezolid compounds have both been shown to be strong antibacterial and antitumor medications, respectively [8]. In addition, a wide range of biological effects is mediated by morpholine derivative ligands and their transition metal complexes, comprising of antidiabetic, antiproliferative, antitumor, anti-inflammatory, anticonvulsant, and antibacterial characteristics [9,10,11,12,13].

Moreover, it is also anticipated that several ligands and metal complexes will be developed that could fit well into DNA binding, cleavage, and interactions with the particular protein that would ultimately serve as the primary cellular target for anticancer medications [14]. Furthermore, numerous biological properties have shown that deoxyribonucleic acid is the main intracellular target of several medicines. Thus, the main focus of several researchers has been on finding and/or creating chemicals with the ability to interact with DNA. Moreover, it is believed that transition metal complexes either bind covalently or non-covalently with DNA [15,16,17]. Furthermore, the electronic structure and molecular characteristics of bioactive compounds in the ground and excited states can be visualized with the use of theoretical calculations and molecular docking-based virtual screening investigations, which also supported understanding the binding mechanism and evaluating the stability of the guest-host molecules [18]. The research work has also been extended due to the fact that complexes (**1b**–**2b**) have strong biological efficiency and may assist in the future creation of powerful new antitumor medications. In addition, the SARS-CoV-2 main protease’s binding characteristics for our substances encourage additional research targeted at confirming the predicted activity and aiding in the fight against existing or upcoming viral pandemics.

## 2. Experimental Section

### 2.1. Materials and Methods

We acquired the essential chemicals and reagents from the Sigma-Aldrich Company. The ligand (**HL**) and its mixed ligand complexes (**1b**–**2b**) were measured by a variety of analytical and spectroscopic studies. The entire dataset was summarized in our earlier reports [4,18], and the additional information was enumerated in the electronic Appendix A.

### 2.2. Assessment of DNA/BSA Interacting Characteristics

#### 2.2.1. Assessment of DNA Nuclease Efficacy

The DNA nuclease characteristics were scrutinized of all substances along with DNA containing H_2_O_2_ and Tris-HCl buffer solution with a pH of 7.4 [19]. The gel layer was discarded from the tank solution after the experiment was finished and put in front of a UV transilluminator, and each band lane was also scrutinized with the control (DNA + H_2_O_2_) [20].

#### 2.2.2. Analysis of DNA-Interaction Characteristics

The DNA-binding experiment was performed by a UV–visible spectrophotometer by raising the DNA concentration from zero to 50 μM to the given concentration of all samples (50 μM) in Tris-HCl buffer with a pH of 7.4 at 25 °C [21,22].

#### 2.2.3. Assessment of Thermal Denaturation Characteristics

The biothermodynamic properties were obtained from absorption titration by a UV–visible spectrophotometer, accomplished with a temperature-controlled sample container in both the presence and absence of the substances. In a 5 mM Tris-HCl/50 mM NaCl buffer solution with a pH of 7.4, each test substance (50 μM) was incubated with CT-DNA (50 μM) in a 1:1 ratio. The test materials were all warmed at a rate of five degrees Celsius per minute to temperatures between 25 and 100 °C, and the changes in absorbance at 260 nm were carefully monitored [23,24,25].

#### 2.2.4. Assessment of DNA Affinity by Hydrodynamic Technique

The hydrodynamic titrations were carried out for all substances along with ethidium bromide (control) at 20, 40, 60, 80, 100 µM concentrations. In the Ostwald viscometer, these substances were also separately treated with the deoxyribonucleic acid solution (100 µM) [26].

#### 2.2.5. Assessment of DNA/BSA Interacting Characteristics by Fluorometric Titration

The DNA/BSA binding properties were acquired from the emission spectral titration with the supporting of the JASCO FP-6300 spectrofluorometer between the 200 and 800 nm regions. It is additional evidence for the complexes’ manner of DNA binding. While DNA (200 µM) was present or absent, we vigilantly monitored the intensity variations between 610 nm and 510 nm during the initial emission and excitation of EB [27]. Correspondingly, the emission titrations for BSA interaction with different concentrations between 2.5 μM and 25 μM for all samples were conducted in a Tris-HCl buffer solution (pH: 7.4) between the regions of 300 nm and 400 nm [28].

#### 2.2.6. Förster’s Theory-Based FRET Computation

As per Förster’s theory, the critical distance of the donor-acceptor system can be determined with the supporting of the FRET approach to assess the binding affinity between BSA-substance systems [29,30,31].

#### 2.2.7. Analysis of DNA Binding Characteristics Using the CV Method

The CV experiment was carried out for free substances at 10 µM at room temperature with a 5 mM Tris-HCl/50 mM NaCl buffer solution (pH: 7.4). While CT-DNA increases (0–10 µM) in each sample solution, shifts in potential, including variations in the anodic and cathodic peak currents, have been monitored [32].

#### 2.2.8. Assessment of BSA Binding Characteristics by UV–Vis Absorption Titration

UV–vis spectral titrations were performed with a 25 μM concentration of BSA at room temperature with a 5 mM Tris-HCl/50 mM NaCl buffer solution (pH: 7.4). When the sample concentrations (0–25 μM) increased in the same BSA concentration solution, the alteration in the absorption band at 278 nm was continuously monitored [33].

### 2.3. DFT and Molecular Modelling Properties

DFT computations were performed to comprehend the electronic structure and reactivity of all test compounds with the support of the Gaussian 09 package [34]. Geometries were fully optimized both in the gas phase and methanol solution using B3LYP, and comparatively later with the CAM-B3LYP and M06-2X functionals. For each complex, the transition metal was treated by the LAL2DZ basis set, while 6-31G(d) was employed for the rest of the atoms. In line with several previous studies, the solvent effect was accounted for in the framework of the polarizable continuum model with the support of the integral equation formalism (IEF-PCM) [35,36]. In addition, docking studies were performed using Autodock Vina to acquire insight into the binding profile of all substances into the active sites of BSA, SARS-CoV-2 main protease (Mpro), and CT-DNA dodecamer [4,37,38,39].

### 2.4. UV–Vis Absorption Titrations for In Vitro Antioxidant Assay

All substances were evaluated for their scavenging capabilities with the support of the UV–vis absorption spectrophotometer at different concentrations of 40, 80, 120, 160, 200, and 240 µM. We carefully monitored the absorbance at 517, 230, 590, and 546 nm while carrying out the antioxidant properties for DPPH, hydroxyl, superoxide, and nitric oxide radical scavenging. In addition, the obtained IC_50_ findings of all test substances were also contrasted with those of standard ascorbic acid [40,41,42].

### 2.5. Assessment of In-Vitro Antimicrobial Properties

In-vitro antimicrobial properties for all compounds were acquired from the agar disc diffusion technique against different selected fungal and bacterial strains [43,44,45]. In addition, the reported inhibition zone findings were correlated with the standard antifungal drugs ketoconazole and amphotericin B, as well as the standard antibiotic medications streptomycin and amikacin.

### 2.6. MTT Cell Viability Assay for Anticancer Characteristics

All samples against the A549, HepG2, MCF-7, and NHDF cell lines were assessed with the supporting of the MTT technique. The acquired data were used to compute the IC_50_ value and contrast it to the gold standard anticancer medication, cisplatin [46].

## 3. Results and Discussions

All of the compounds are highly coloured, faintly hygroscopic, and have high solubility in CH_3_OH, C_2_H_5_OH, CHCl_3_, and DMSO at room temperature. The evaluated analytical results and structural characteristics are deposited in the Appendix A.

### 3.1. Synthetic Process and Properties

All observed analytical results, structural characteristics, as well as crystallographic data for ligand (**HL**) and its mixed ligand complexes (**1b**–**2b**) (Figure 1 and Appendix A), are deposited in the Appendix A.

### 3.2. DNA/BSA-Binding Properties

Typically, it is recommended to restrict the growth of tumour cells by preventing the reproduction of DNA that has been damaged or broken due to binding or cleavage mechanisms. It deals with the static binding mode between the substance and BSA.

#### 3.2.1. Analysis of DNA Cleavage Characteristics

DNA cleavage is a typical and essential process for maintaining cell viability. During DNA replication and transcription, topoisomerase enzymes correct topological issues, and various nucleases take part in repair mechanisms and DNA degradation, which is one of the distinguishing features of apoptotic programmed cell death. Similarly, several antitumor drugs have the potential to cleave the DNA by inducing apoptosis, which eventually leads to cancer cell death. In this case, ligand and its metal complexes are involved in causing oxidative cleavage in the presence of H_2_O_2_ by the gel electrophoresis method, and the observed findings are revealed in Figure 2 (Lanes: 1–4). Due to their effective DNA interacting capacity, the complexes had higher cleavage efficiency than the control. No discernible DNA cleavage was seen at a long exposure time for the control (DNA alone) (Lane: 1; DNA + H_2_O_2_) and free ligand (**HL**) (lane: 2), which was also monitored under the H_2_O_2_ environment. Lanes: 3 shows that the complex (**1b**) demonstrates complete DNA cleavage. Similarly, Lane: 4 reveals that complex (**2b**) partially cleaves DNA. In addition, the performance of the band reduction in the lanes was revealed in the agarose gel (Figure 2). However, lanes: 2 shows that free ligand (**HL**) has no considerable cleavage efficiency in the series of metal complexes. DNA cleavage may occur via abstracting an H-atom from sugar units and releasing particular residues from transformed sugars, depending on the position from which the H-atom is eliminated. Meanwhile, the hydroxyl radicals strongly involved in DNA cleavage are followed by oxidative mechanisms. However, the concentration of the complex, the type of metal ion, and the presence of H_2_O_2_ as exogenous reagents are the main variables that influence the extent of the DNA cleavage mechanism [47]. If complexes have weak hydrogen abstraction, they expose less cleavage activity. The redox characteristics of the core metal ions, where zinc is redox-inactive but other metal ions are redox-active, are one reason for the complexes’ varying reactivity. Therefore, activity in complex (**2b**) indicates incomplete cleavage. However, the complexity of the complexes and their affinity for DNA do not adequately account for the relative efficacy of the various metals at cleaving DNA [48]. Furthermore, ROS include superoxide (O_2_^•−^), hydrogen peroxide (H_2_O_2_), Hydroxyl (OH•), hydroperoxyl (ROOH), carboxyl (ROO•), hypochlorous (HOCl), singlet oxygen ^1^O_2_, and ozone (O_3_), which play essential roles in living systems [49]. ROS has the ability to alter cell function as well as affect the genesis of cancer at several levels. In this case, ^●^OH can attack DNA, proteins, and lipids due to its high reactivity among ROS. The hydroxyl radical plays an imperative role in free radical-mediated hazardous reactions. Free radicals are essential in the redox regulation of many cell signalling pathways and proper cellular functions. Prior to it being realized that free radicals are only generated in living systems, superoxide (O_2_^●−^) was believed to be a typical cellular metabolite. It was then realized that more dangerous radicals could potentially be produced via the Haber-Weiss process. The combination of O_2_^●−^ and H_2_O_2_ may produce powerfully reactive ^●^OH radicals [50]. Nevertheless, as per the Fenton/Haber-Weiss mechanism, it is suggested that it is capable of vigorous nucleolytic cleavage by chemical substances in an oxidizing agent (H_2_O_2_) environment [51]. According to this mechanism, the complexes acted as excellent biocatalysts for the formation of the diffusible ^●^OH free radicals from hydrogen peroxide. Additionally, ^●^OH free radicals abstract the H-atom from the sugar fragment of the DNA base pair to generate sugar radicals. Concerning the location of the hydrogen atom, it rapidly induces the hydrolytic nuclease activity at the sugar-phosphate backbone [52]. The rapid migration of deoxyribonucleic acid can lead to the open circular form’s transformation into a linear form. Moreover, EDTA facilitates the formation of highly reactive diffusible ^●^OH and anions via the Fenton or Haber-Weiss processes and prevents metal ions from interacting with DNA due to the formation of an EDTA-Metal system. The diffusible hydroxyl free radicals also stimulate the abstraction of the H-atom from the sugar part of the deoxyribonucleic acid base pair to generate sugar radicals along with the formation of an adduct with nucleotides. Therefore, DNA cleavage occurs owing to the assault of a diffusible ^●^OH on deoxyribonucleic acid base pairs under the metal complex environment. The complex serves as an effective catalyst for the production of ^●^OH from hydrogen peroxide according to the Fenton mechanism (Appendix A) [53].

#### 3.2.2. Assessment of DNA Binding Properties Using UV–Visible Absorption Titration

UV–visible spectral titration is an excellent technique for determining the interaction modalities between DNA and metal complexes. The intensity of the intercalative binding is determined by the percentage of hypochromism or hyperchromism in the MLCT band [54]. While raising the concentration of deoxyribonucleic acid solution to the fixed concentration of the compound, the optical density of the band gradually decreased (hypochromism) with a red shift being observed [55], as shown in Figure 3. In addition, a substance that intercalates into DNA often attain in hypochromism with or without a red or blue shift because of the strong stacking interaction between the planar aromatic chromophore and the DNA base pairs in the intercalative mode [56]. While this is happening, the red shift coincides with a reduction in the energy gap between HOMO and LUMO after full binding of the compound with DNA [57]. Moreover, DNA binding to metal complexes presents the essential information regarding conformational change, the effectiveness of the DNA-substance binding and the negatively charged phosphate on the deoxyribonucleic acid is neutralized by exterior contact and intercalation through interactions of π–π stacks. Several literatures reported that the following four kinds of non-covalent interactions play an essential role in biomolecular interaction properties. Here, (i) involves a negatively charged phosphate fragment as a result of electrostatic interaction, (ii) influences of weak van der Waals force attraction/H-bonding, (iii) interaction of a functional moiety with the grooves (major/minor) of the double-stranded DNA as a result of a molecule sticking due to general attraction, or as a result of water or H-bonding expulsion, etc. (iv) The stacked base pairs of natural DNA are intercalated by hydrophobic forces. Nevertheless, substances are engaged in the reaction because of quinine’s preferred N-7 position and adenine’s N-3 location in DNA. The DNA base pairing may also be prevented due to miscoding. All complexes (**1b**–**2b**), including free ligand (**HL**), were measured in the presence and absence of DNA in buffer solution with a pH of 7.4 at 25 °C (Figure 3). The results were also included in Table 1 and Appendix A. In this case, all substances were exposed to two prominent electronic absorption bands of about 260 nm and 334–342 nm, consequent to the π–π* transitions of the phenyl chromophore and MLCT, respectively. While the amount of DNA in each compound rises, the interaction of the chemical substance with DNA base pairs generates noticeable alterations in the intra ligand charge transfer bands’ strength and wavelength. The hypochromic shift of all compounds was observed in the range of 46.88–56.88% with 4–7 nm red shifts, which occurred due to diminishment in the π–π* transition energy and the half-packed electrons of bonding orbitals. In contrast, it would be possible for electrostatic interaction if the complex-DNA adduct exhibits hyperchromism with a hypsochromic shift [58,59]. Using Wolfe-Shimmer Appendix A, the observed K_b_ values for all samples were estimated from the linear regression plot of [DNA]/(ɛ_a_–ɛ_f_) vs. [DNA] and the K_b_ results for each sample were in the subsequent sequence: (**1b**) > (**2b**) > (**HL**). Moreover, the overall observed ${\Delta G}_{b}^{{}^{\circ}}$ values in all cases were in the range of −22.50 to −25.12 kJmol^−1^ (Table 1), which also indicates that the compounds spontaneously intercalate to DNA [60,61]. However, the complex (**1b**) exhibited excellent binding potency compared to others. It is concluded that the morpholine-linked ligand’s co-planarity and 1,10-phenanthroline aromatic system complexation with the metal centre promote the complex’s ability to infiltrate DNA base pairs smoothly, and large aromatic systems may also assist the complex to penetrate the phosphate backbone’s core deeply, and those substances may permit the complex to freely penetrate deep into the deoxyribonucleic acid base pairs. In addition, the isosbestic point is found at 286 nm for free ligand, which also suggests that DNA and ligand establish a dynamic equilibrium and further concludes that compounds spontaneously intercalate into DNA and further attain the hyperchromic effect after isosbestic point (Figure 3). The Wolfe–Shimmer (Appendix A) [62], Benesi–Hildebrand (Equations (S3) and (S4)) [63] and Sakthi–Krause Appendix A were supported to evaluate K_b_ results for all samples, and all equations were included in the Appendix A. The K_b_ values were measured using the Wolfe-Shimmer Appendix A by methods I and II, from the linear regression plots of [DNA]/(ɛ_a_ − ɛ_f_) vs. [DNA] M^−1^ and (ɛ_b_ − ɛ_f_)/(ɛ_a_ − ɛ_f_) vs. 1/[DNA] M^−1^ (Appendix A). The Benesi–Hildebrand binding constant (K_b_) values were measured using Appendix A by methods I and II, from the linear regression plots of [1/(A_x_ − A_0_)] vs. {1/[DNA]} M^−1^ and [(A_max_ − A_0_)/(A_x_ − A_0_)] vs. {1/[DNA]} M^−1^ (Appendix A). The K_b_ values were estimated using Sakthi–Krause Appendix A by methods I and II, from the linear plots of [(A/(A_0_ − A)] vs. {1/[DNA]} M^−1^ (Appendix A) and {1/[DNA]} vs. log [(A/(A_0_ − A)] M^−1^ (Appendix A). In addition, A_0_ and A are represented as the absorbance intensity values in the absence and presence of the [DNA], respectively. The Van’t Hoff Equation (S7) was allowed to obtain the ${\Delta G}_{b}^{{}^{\circ}}$ values for DNA interaction, and the Equation (S8) was supported to measure the % of chromicity of all substances (ESI-2a). The overall measured K_b_ findings for all substances were in the subsequent sequence: (**1b**) > (**2b**) > (**HL**). In these cases, the observed ${\Delta G}_{b}^{{}^{\circ}}$ values were reported in the range of −24.58 to −25.12, −24.47 to −24.65 and −22.50–−23.92 kJmol^−1^, respectively (Table 1). Nevertheless, the complex (**1b**) had the highest DNA binding efficacy among others. In addition, the findings of DNA cleavage, emission, hydrodynamic, and CV measurements are well supported by the preceding observation.

#### 3.2.3. Assessment of Thermal Denaturation Characteristics

DNA denaturation is one of the prime causes of a number of chronic diseases, hereditary disorders, and a reduction in the ability of DNA repair to work properly. A greater risk of cancer and illnesses that hasten aging and mutations linked to a lack of DNA repair can cause Fanconi’s anemia, blood syndrome, and xeroderma pigmentosum. Most often, thermal denaturation tests can be recorded with an absorption spectrophotometer by examining the absorbance at 260 nm [64]. The biothermodynamic properties were further supported in order to determine the ability of stabilization of the double standard DNA, and it offers details on the structural alterations, the degree of the DNA-compound system, the external binding-mediated neutralization of the phosphate charges on DNA, and the stacking interactions, all of which work together to raise the DNA’s melting point [65]. Moreover, small molecules are involved in the reaction due to the DNA’s preferred N-7 site for guanine and N-3 site for adenine. Therefore, it is possible to block the deoxyribonucleic acid double helix, which causes miscoding of deoxyribonucleic acid. In this case, it is observed that complex -DNA adducts have a higher melting temperature compared to free DNA. Compound-bound deoxyribonucleic acid is more challenging to melt compared to deoxyribonucleic acid alone because it is involved in powerful intercalation binding with DNA. The Van’t Hoff (Appendix A) and Gibbs Helmholtz Appendix A were supportive in evaluating the biothermodynamic parameters, which are enclosed in Table 2. The evaluated T_m_ values for deoxyribonucleic acid alone were 68 ± 2 °C in the same experimental setting, and the T_m_ values of the DNA-compound adduct were obtained in the following sequences: 80.5 °C (**1b**) > 78.5 °C (**2b**) > 74 °C (**HL**) and melting temperature changes (ΔT_m_ °C): 12.5 (**1b**) > 10.5 (**2b**) > 6 (**HL**). When ΔT_m_ > 10 °C, the described biothermodynamic properties indicate the possibility of an intercalation mode of the mechanism between compound and deoxyribonucleic acid, with the exception of the free ligand. In the case of ΔT_m_ < 10 °C, which reveals the electrostatic or groove binding mode [66,67,68,69]. Additionally, the complex-DNA adduct’s reported negative binding free energy was less than compared to the ligand-DNA binding energies, which attributes the complexes (**1b**–**2b**) spontaneously intercalating DNA (Figure 4a,b and Table 2). The influencing factors between test compounds and DNA mostly depend on the type of interaction mode. Due to the driving forces, H-bonds, weak Van der Waals forces, and electrostatic modes of binding all occur when enthalpy is favourable. Hydrophobic forces induce binding while entropy is favourable. In contrast, the loss of structural degrees of freedom leads to undesirable entropic changes. As per Ross and colleagues, the findings for ΔH° and ΔS° can alternatively be derived in the subsequent favourable sequence: ΔH° > 0 and ΔS° > 0, which attribute intercalation to hydrophobic forces of attraction. If ΔH° < 0 and ΔS° < 0, which involved weak Van der Waals forces of attraction and H-bonding interactions. On the other hand, if ΔH° < 0 (or) ΔH° ≈ 0 and ΔS° > 0, which indicates the electrostatic modes of binding possible between DNA and compounds [70]. The measured values for all samples were exposed to the favourable sequence ΔH° < 0 and ΔS° < 0, which is assumed to be due to weak Van der Waals forces of attraction and H-bonding between DNA and chemical substances. However, they lose the ability to rotate and translate, interfere with counterions and hydrophobic forces in compound-DNA adduct, and may result in exothermically active negative signals of ΔS° and ΔH°. Furthermore, it is widely acknowledged that hydration and the formation of the compound-deoxyribonucleic acid adduct system via the counter ion liberating mechanism are highly dependent on hydrophobic forces of attraction. As a result, the higher negative results of ΔH° and ΔS° for all substances that interacted with DNA were observed in the experiment [71]. According to the Ross and Subramanian mechanism for protein/DNA-complex interactions, it obviously reveals that the complexation of the metal centre with the morpholine fused primary aromatic and 1,10-phenanthroline secondary aromatic planar ring systems stimulates the silky penetration of the complex within deoxyribonucleic acid base pairs owing to π–π stacking interactions. Additionally, in the complex-DNA adduct, a number of non-covalent molecular interactions, including dipole-dipole interaction, weak Van der Waals forces of attraction, formation of H-bonds, and electrostatic forces of attraction, may be present while the complex is positively charged and encompass stacking interactions as per Manning and Record’s polyelectrolyte hypothesis [72].

#### 3.2.4. Assessment of DNA Binding Affinity Using Viscometric Techniques

Viscosity measurement is widely recognized as an effective technique for determining the binding affinity between test compounds and DNA. Most intercalation agents are responsible for increasing the viscosity of DNA [73]. In contrast, the viscosity of DNA is slightly influenced by electrostatic or groove bindings. While small aromatic planar molecules intercalate between adjacent base pairs via hydrophobic forces, it leads to an in-crease in the contour lengthening of the DNA, which is responsible for increasing viscosity. The increased viscosity of DNA solution may be related to a particular intercalation binding mechanism since the viscosity of DNA solution is highly sensitive to changes in the contour length of DNA [74]. In addition, the degree of viscosity may depend on its affinity for deoxyribonucleic acid. While rising the affinity of DNA with test compounds, the degree of viscosity also increased. Generally, if the viscosities of DNA in the Tris-HCl buffer solution do not alter with the rising concentration of compounds, this indicates that the binding mode of DNA with complexes may be groove bindings due to electrostatic, partial, and non-classical interaction modes. Therefore, it is clearly noted that the final viscosity of the DNA solution remains unchanged. In this case, the DNA binding properties of all test compounds were further confirmed by the hydrodynamic techniques because they are susceptible to DNA contour length changes, which means that the average distance between each monomer (cl = 0.338 nm/bp for B-form DNA). It was observed that the absolute viscosity increased consistently along with the concentration of each substance at the fixed DNA concentration. As a result of the strong binding mode of intercalation, the contour length of the double-helix DNA rises [75]. In addition, the affinity interaction and their slope values were observed from the relative specific viscosity (*η*/*η*_0_)^1/3^ plotted straight line verses [Compound]/[DNA], and absolute specific viscosity of DNA in the absence or presence of test compounds was evaluated with the supporting of Appendix A. In the experiment, it was clearly noted that the slope values for all samples also increased due to the rising binding affinity. The evaluated slope findings were found in the following order: (**EB**) 1.286 > (**1b**) 1.085 > (**2b**) 0.840 > (**HL**) 0.462 (Appendix A and Table 3). However, complex (**1b**) exhibited superior binding affinity among others and was substantially smaller than **EB**. Due to the existence of 1,10-phenanthroline and the morpholine-fused aromatic planar systems, the compounds can interact with deoxyribonucleic acid robustly via intercalation. The results were in excellent agreement with the observed findings of UV–Visible spectral properties.

#### 3.2.5. Assessment of DNA/BSA Binding Characteristics Using Emission Titration

In general, the fluorescence of ethidium bromide (EB) is quite weak in aqueous solution, but when it is bound to deoxyribonucleic acid, the fluorescence intensity rises. However, they play a well-known imperative intercalator and are more helpful in distinguishing the binding strength of the non-fluorescence test substances. Furthermore, in the absence and presence of rising quantities of each test compound, the fluorescence emission spectra of the EB-DNA adduct was observed at 610 nm. When the complex concentration rises, the fluorescence intensity of the EB-bound DNA complex diminishes owing to the displacement of EB from CT-DNA. A notable reduction in the fluorescence emission intensity at 610 nm is observed (Figure 5 and Table 4). The photoelectron shift from DNA’s guanine base to the excited states may be the cause of the frequency of quenching in the emission of the chemical substance by DNA. Additionally, after each compound was added to EB, no additional peaks were noted, which shows that EB did not cause any quenching of its free fluorescence emission and proves that the compounds did not interact with EB. The band’s intensity dropped dramatically as increasing amounts of each test substance were added to the fixed concentration of the EB-DNA adduct, demonstrating the capability of the investigated compounds to displace bound EB from DNA [76,77,78,79,80]. In this investigation, the addition of sample concentrations (0–240 μM) to the solution of DNA–EB results in a notable reduction in the intensity of emission at 610 nm (Figure 5 and Table 4). As a result of intense intercalation, the complexes displace the EB from the DNA–EB adduct, causing the emission intensity to drop. The photoelectron shift from the DNA’s guanine base to the excited states may be the cause of the frequency of quenching in the emission of chemicals by DNA. Therefore, EB can be utilized as a fluorescent probe in the competitively interacting experiment.

Additionally, the fluorescence intensity of bovine serum albumin was monitored at 350 nm (λ_ex_ = 278 nm) during fluorescence titrations. When increasing the sample concentrations, owing to static quenching in the ground state bovine serum albumin–compound adducts, the BSA intensity diminishes dramatically (Appendix A). Meanwhile, it is noted that there is no shift in the fluorescence spectrum. Furthermore, it is assumed that compounds may interact with BSA and that the fluorophores of BSA are not clearly exposed to a shift in polarity [81,82]. Additionally, the Stern-Volmer Appendix A were employed to analyse the data (Appendix A and Table 4), and the kq values for DNA and BSA binding were acquired in the range of 1.1636–4.0303 × 10^12^ and 2.6390–9.564 × 10^12^ mol^−1^s^−1^, respectively. They are also much greater than the collision quenching constant value (2.0 × 10^10^ mol^−1^s^−1^). Therefore, it is assumed that the static quenching process was brought on by adduct construction between the compound and bovine serum albumin rather than a dynamic collision. However, fluorescence spectroscopy is generally plagued by the inner filter effect (IFE), which causes particularly disturbing spectral analysis. The energizing ray is attenuated due to the highly concentrated solution sample. As a result, strong fluorescence is only seen on surfaces facing the excitation beam. The fluorescence intensity is reduced as a result of an inner filter effect generated by some chemicals’ absorption of the excitation/emission wavelengths in the UV province. The observed values of the absorption wavelength of ligand (**HL**) and complexes (**1b**–**2b**) in the range of 334–336 nm, the bovine serum albumin excitation wavelength of 278 nm and the emission wavelength of 350 nm were supported to assess the effect of IFE in this approach (Figure 5). Appendix A was employed to solve IFE in this instance as well (Appendix A) [83]. The emission intensities of EB that interacted with DNA at 610 nm and those of bovine serum albumin at 350 nm exhibited a distinctly reducing movement with increasing concentrations of the test compounds when the IFE was resolved, indicating that after being replaced with the substances, a few ethidium bromide molecules were released into solution, which caused ethidium bromide’s fluorescence to be quenched. Moreover, in all cases, the observed R^2^ values for the linear plots of F_0_/F vs. [Q] and log (F_0_ − F)/F vs. log [Q] by Stern–Volmer (SV) method I and II were nearly 1 (*p* ≤ 0.05), which is also more supportive to linear regression model and one of the significant factors for measuring the impact of the inner filter effect (Appendix A).

Additionally, no emission spectrum shifting was seen the BSA-compound adduct, indicating that ground state BSA-compound systems formed on account of a static quenching mechanism. It was further evaluated that BSA might interact with complexes and that the polarity of BSA’s fluorescence did not vary noticeably with complex titration. These findings, which were reliable with the UV–vis spectral data, can be interpreted as the complexes’ intercalation mode between deoxyribonucleic acid base pairs and BSA. The following Stern-Volmer Appendix A were employed to determine the K_SV_, K_q_, and n values Appendix A. The K_SV_ values were measured from the linear regression plot of F_0_/F vs. [Q] by the SV method I (Appendix A). Appendix A was employed to evaluate the n and K_ass_ values [84]. Similarly, K_app_ (apparent binding constant) values for all samples were estimated using Appendix A. K_ass_ and n were measured from the linear regression plot of log (F_0_ − F)/F vs. log [Q] by SV method II with the supporting of Appendix A and Table 4). In this case, ε values for all substances were observed on the linear regression plot of fluorescence emission intensity vs. [compound] with the supporting of the Beer-Lambert law equation (A = εcl) Appendix A is supported to evaluate K_app_ value using K_EB_ = 10^7^ M^−1^ at 50 µM concentration, and measured the sample concentrations for all cases using the Beer–Lambert law equation. The complex concentration’s IC_50_ finding was estimated at a 50% diminution in EB’s intensity of emission. The Lineweaver-Burk Appendix A and Scatchard analysis Appendix A are allowed to expand the observations and validate the binding affinities [85,86] and the observations are also compared with the Stern–Volmer method Appendix A. The overall measured binding constant values (K_SV_, K_ass_, K_app_, K_LB_ and K_SC_ × 10^4^ M^−1^) for all samples were in the subsequent order: (**1b**) > (**2b**) > (**HL**) (Appendix A). The n values for DNA/BSA binding acquired from the Stern–Volmer Appendix A and the Scatchard Appendix A were in the range of 0.9760–1.0880 and 0.9090–1.0966 for all compounds, respectively (Table 4), which also obey the neighbour-exclusion principle. In these cases, complex (**1b**) has shown a better binding affinity than other compounds owing to the value of n being nearly equal to one. Consequently, it is proposed that the complexes contain both a 1,10-phenanthroline ring planar system and an aromatic ring system linked with morpholine. They can effectively interact with deoxyribonucleic acid via intercalation. Additionally, the values of the fluorescence quantum efficiency (P) ratio for the deoxyribonucleic and bovine serum albumin-complex adducts were 0.0900–0.1280 and 0.2250–0.3220, respectively, which were measured from linear regression plots of F/F_0_ vs. 1/[DNA] and 1/[BSA], respectively (Appendix A, and Table 4). The findings and those from the viscosity, electrochemical titration, and UV–Vis spectral properties were in good accordance with the outcomes.

#### 3.2.6. Förster’s Theory-Based FRET Computation

FRET (Fluorescence Resonance Energy Transfer) can be employed to distinguish the relative orientation and closeness of fluorophores [87]. The process happens while there is a large overlapping of the acceptor’s (compound) electronic spectrum with the donor’s emission spectrum (BSA). The “r” is represented as the Förster distance, which is measured between the substance and bovine serum albumin, and the distance has to be between the prescribed limits of 2 and 8 nm in order for energy transfer to occur. Fluorescence is quenched owing to energy being transmitted from the excited state of bovine serum albumin to chemical substances. As a result of the FRET analysis, the “r” findings were obtained in the range of 2.6685–2.8257 nm (Table 5 and Figure 6). It also shows that energy will probably be transmitted with a high likelihood from bovine serum albumin to the compounds. FRET happens while the chromophore’s absorption spectrum and the fluorophore’s emission spectrum overlap. The following conditions have a major impact on the FRET’s effectiveness: (i) the distance (r) should be between 2 and 8 nm for energy transfer, (ii) there is a large overlap of the emission spectrum of biomolecules (donors) with the electronic absorption spectrum of acceptors (substances), and (iii) the bovine serum albumin and substance transition dipoles are correctly oriented. Bovine serum albumin transmits excitation energy to a compound during FRET without emitting a photon from the previous molecule system. It involves a number of molecular electronic excited states that are dependent on distance. The evaluation of the r values between micromolecular and biomacromolecular systems is facilitated by FRET (Figure 6). Appendix A can be employed to estimate the efficacy of energy transfer (E) as per the FRET theory (Appendix A). Basically, the K^2^ values were found in the range from 0 to 4, and energy can be transferred from the BSA to the compound when electrons are transferred between the two molecules. For parallel transition dipoles that are aligned, K^2^ is equal to 4, which denotes maximal energy transfer. When the orientation of the dipoles is perpendicular to one another, K^2^ is equal to 0, which denotes very weak energy transfer. When the relative orientation of the dipoles is at random, K^2^ is attained to be equal to 2/3. Here, n is denoted as the average refracted index of the medium, Φ is represented as the fluorescence quantum yield of the BSA, and Appendix A is helpful to measure the normalized spectral overlap integral (J) for overlapping the emission spectrum of the BSA with the electronic spectrum of the compound. The molar absorption coefficient (ε_A_) and the fluorescence emission intensity were both adjusted to the unit area on the wave number scale. It is imperative that J, after being normalized, be independent of the real size of ε_A_. The following variables for the complex-BSA interaction are determined using Appendix A, n = 1.36, Φ = 0.15, E = 0.2769–0.3462, J = 0.6852–0.9460 × 10^−14^ cm^3^ L mol^−1^, R_0_ = 2.3673–2.4980 nm, r = 2.6685–2.8257 nm, k_ET_ = 4.7330–5.8443 J/s and B = 4999.08–5650.81 mol^−1^ cm^−1^ (Table 5). The observed findings of R_0_ and r between BSA Trp213 and the interacted compound were substantially smaller than 8 nm and their relationships for all compounds are found in the following sequence (Table 5): 0.5 R_0_ (1.1836–1.2490) < r (2.6685–2.8257) < 1.5 R_0_ (3.5510–3.7470). This implied that there was a potent possibility that the test compound and BSA exchanged non-radiative dipole-dipole energy, which was consistent with a static quenching mechanism. This result proved that the binding adhered to the conditions of Förster’s energy transfer theory. In this case, Φ is denoted as the quantum yield, which is defined as the dimensionally invariant ratio of photons emitted to photons completely absorbed by a fluorophore, and it serves as a dynamic tool for estimating the effectiveness of fluorescence emission in correlation to all other channels of relaxation. In addition, τ is denoted as the lifetime of fluorescence emission of the biomolecule and is described as the inverse of the entire degradation rate τ = 1/(k_r_ + k_nr_). The fluorophore’s radiative lifetime is represented as τ_0_ = 1/k_r_. The values of τ and Φ are associated with the Appendix A. In addition, quenching occurs while a BSA’s ground or excited states come into contact with a compound in the solution. This is also reflected in the diminishing emission intensity, which is divided into the two main categories of dynamic and static quenching. While in an excited state, BSA binds with the substance during a dynamic or collisional quenching mechanism, whereas the radiation-free deactivation of BSA results in the ground state. Therefore, the concentration of the quenching compounds affects dynamic quenching. The τ and Φ values for BSA, diminishes with raising the compound concentration. Conversely, static quenching diminishes emission without changing the excited state τ or Φ, and quenching can be divided into two main categories based on the excited-state lifetime of the fluorophore. Additionally, k_q_ [Q] is included in the Appendix A and the Φ finding for the bovine serum albumin-compound adducts system is measured by the Appendix A. FRET requires an interaction between the emission and the electronic transition dipole moments of the bovine serum albumin and test compound, respectively, due to the non-radiative transfer of excitation energy from a fluorophore to a chromophore [88]. In this case, k_ET_ is denoted as the energy transfer rate, which is dependent on not only the overlapping spectrum of emission of the BSA, and the absorbance of the compound, but also the Φ values of the BSA, K^2^, and r, etc. The k_ET_ values for all substances were estimated by Appendix A and the Förster radius is the distance at which resonance energy transfer is 50% efficient (R_0_) [89]. In addition, the brightness of BSA depends on the capability of a test compound to absorb light and the Φ value, which is calculated by the Appendix A. Chemical compounds with high absorbance have higher values for ε and Φ, which also promote effective emission.

#### 3.2.7. Analysis of DNA Binding Characteristics Using the CV Method

CV has proven to be a very sensitive analytical technique in order to identify alterations in the redox behaviour of metallic components in the presence of physiologically significant chemicals [90]. The consecutive decrease in peak current and progressive peak potential shift in the direction of positivity at both cathodic and anodic peaks attest to the drug’s altered voltammetric response, suggesting that the compound intercalated into deoxyribonucleic acid. The examined reduction of the anodic and cathodic currents implies that the complexes have bound to the massive, gradually diffusing deoxyribonucleic acid molecule [91]. In addition, the CV approach is one of the most important tools for examining the DNA-complex adduct’s binding mechanism. The CV properties of all test samples in the presence and absence of deoxyribonucleic acid were executed at a scan rate (*v*) 0.1 Vs^−1^ with a potential range of +2 to −2 in a Tris-HCl (5 mM)/NaCl (50 mM) (pH: 7.2) solution. The M^1+^/M^2+^ redox couple is caused by complexes that reveal a single anodic and cathodic peak. The complex’s reaction with the glassy carbon electrode surface was shown to be a one-step, one-electron, quasi-reversible redox process since the (I_pa_/I_pc_) ratio values of the redox couple were nearly unity, which is also supported by the change in peak potential separation (Ep > 0.0591 V) [92,93,94] (Appendix A and Table 6). While the substances often bind to deoxyribonucleic acid through intercalation, the peak potential shifts in a positive direction. On the other hand, while the substances bind to deoxyribonucleic acid via minor or major grooves or electrostatic attractions, the peak potential shifts occur in a negative direction. In this case of ligand (**HL**) and its complexes (**1b**–**2b**), due to the consistent movement in the positive direction caused by the increment of deoxyribonucleic acid, the binding mode has been described as mainly intercalation in the compound-DNA adduct (Appendix A), and it is also attributed to 1,10-phenanthroline’s presence and morpholine- fused aromatic planar systems in mixed ligand complexes, which can create inclusion through intercalation owing to hydrophobic and π–π stacking interactions in the deoxyribonucleic acid base pairs. It is also supported by the evaluated outcomes from UV–Vis spectral, emission titration, viscometric, and biothermodynamic properties.

Furthermore, the binding constants, binding site size (S) (bp), and the ratio of binding constants (K^1+^/K^2+^) for M^1+^/M^2+^ couple systems further confirmed the binding affinity via intercalation. In addition, the subsequent Appendix A are supported to determine the above parameters [95,96]. Appendix A is acquired from the Stern–Volmer Appendix A and Table 6) Appendix A. The values of S and K_b_ are estimated from Appendix A using S = (intercept/4)^1/2^ and (K_b_ = 2S (Slope/intercept), respectively, from the linear regression plot of (C_p_/C_f_) versus [DNA] by method II [97,98,99] (Appendix A and Table 6). In addition, base pair sites with a molecule of the compound are referred to as “binding site size” (S), and the evaluated S values were found in the range from 0.2480 to 0.4460 bp (Table 6). In general, if S value is less than one, which denotes stronger binding through intercalation, and S value is greater than one, this suggests the possibility of the modes of groove binding or electrostatic interactions [100,101,102,103,104]. The S value also suggests that there should be one binding site for every two base pairs, representing that complex (**1b**) has exposed higher binding efficiency than others owing to robust binding affinity with DNA through intercalation. It is therefore stated that a compound or medication exhibits high binding affinity when it occupies a single binding site. Meanwhile, when the number of site sizes increases, the drug-DNA adduct exhibits low binding affinity [105]. In the Nernst Appendix A for the galvanic cell, $E_{s}^{{}^{\circ}}=\left(E_{b}^{{}^{\circ}}-E_{f}^{{}^{\circ}}\right)$, the formal electrode potentials of the M^1+^/M^2+^ couple in their bound and free forms are E_1/2_ or $E_{b}^{{}^{\circ}}$ and $E_{f}^{{}^{\circ}}$, respectively. As a result of the variable binding state [M^1+^/M^2+^] and a change in the redox potential to a greater positive direction, and a decrease in both the anodic and cathodic peak currents is made possible by the delayed mass transfer of complexes that bind with DNA fragments. In particular, the equilibrium of M^1+^/M^2+^ is influenced by electrostatic or hydrophobic interactions. However, the $E_{s}^{{}^{\circ}}$ values of M^1+^/M^2+^ for all substances were observed to be positive values (Table 6), which suggests that the compounds’ strong hydrophobicity makes their interactions with deoxyribonucleic acid through intercalation more favourable. On the other hand, if the value of $E_{s}^{{}^{\circ}}$ is negative, this indicates that the substance interacts more favourably with DNA through electrostatic interactions. K^1+^ and K^2+^ are represented as binding constants for the binding states of the +1 and +2 chemical substances to deoxyribonucleic acid, respectively. The “n” is represented as the number of electron transfers, which is equal to one. With the aid of Appendix A, the linear regression plot of log (1/[DNA]) versus log (I/I_0_ − I) (method 1) was employed to determine the ratio of K_[red]_/K_[oxi]_ for reduction and oxidation processes, which was also estimated using the Nernst Appendix A and Table 6). Generally, the DNA-compound adduct is assigned the grooves or electrostatic binding interactions when the ratio value of [K^1+^/K^2+^] is equal to one. When the ratio value is less or greater than one, it demonstrates that the mode of intercalation binding could take place in the DNA-compound system owing to hydrophobic forces of attraction [106,107]. In this case, the following mechanism led to the latter finding in the compounds-deoxyribonucleic acid systems (Table 6).


$$\begin{array}{c} \;\;\;{\mathrm{ML}}^{2+}\;\;\;\;+\;\;e^{-}\;\Leftrightarrow \;{\mathrm{ML}}^{+}\;\;\;\;\;\;\;\;E_{f}^{0}\\ K_{2+}⥮\;\;\;\;\;\;\;\;\;⥮\;\;K_{+}\\ \;\;{\mathrm{ML}}^{2+}-\mathrm{DNA}\;+\;e^{-}\Leftrightarrow \;{\mathrm{ML}}^{+}-\mathrm{DNA}\;\;E_{b}^{0} \end{array}$$


In Appendix A, I_po_ and I_p_ are represented as the peak currents of the complexes (**1b**–**2b**) in the absence and presence of deoxyribonucleic acid. By using method III, the K_b_ value was attained from the reciprocal of the slope in the linear regression plot of Ip^2^ versus (I_po_^2^ − I_p_^2^)/[DNA] (Appendix A and Table 6). In these cases, complex (**1b**) showed greater binding effectiveness than others owing to its robust binding affinity with deoxyribonucleic acid through intercalation. As a result, it is proposed that the present complexes consist of an aromatic planar linked with a morpholine moiety as well as 1,10-phenanthroline aromatic planar systems that may strongly interact with deoxyribonucleic acid via intercalation, and it is also validated by the finding of the diffusion coefficient (D_0_) of the substance alone, and the deoxyribonucleic acid-bound substance with the aid of the subsequent quasi-reversible Randles–Sevcik Appendix A [108]. It is also measured with the help of the Bard–Faulkner equation [α = 47.7/(EPa − EP/2) (Table 6) [109]. On adding DNA to all test compounds, the anodic/cathodic peak currents of the M(I)/M(II) reduced due to a diminishing of D_0_. It is obviously suggested that the evaluated values of D_0_ of deoxyribonucleic acid-bound compounds (_b_D_0_ = 2.5809 − 4.0688 × 10^−5^ cm^2^ s^−1^) were less than the free test compounds (_f_D_0_ = 2.8570 − 4.4035 × 10^−5^ cm^2^ s^−1^) (Table 6). The values of D_0_ for all samples in the absence and presence of deoxyribonucleic acid at scan rates of 0.01–0.3 V/s were measured from the linear regression plots of _f_Ipa vs. *v*^1/2^ and _b_Ipa vs. *v*^1/2^ using the Appendix A [110,111,112] (Appendix A and Table 6).

### 3.3. Evaluation of BSA Binding by UV–Visible Spectral Titration

The titration was executed for BSA in the presence and absence of substance in Tris-HCl (5 mM)/NaCl (50 mM) (pH = 7.2) solution (Figure 7 and Table 7). Quenching typically occurs in either a static or dynamic phase. The static quenching mechanism only involves the formation of a BSA-substance in the ground state, but a dynamic quenching mechanism involves the temporary presence of the excited state, which brings the BSA and the compound into close proximity. In addition, the dynamic quenching mechanism has no effect on the absorption spectrum, as it only affects the excited state [113]. The BSA’s electronic intensity was found at 278 nm. When the test sample concentration rises, the absorbance values also rise accompanied by the blue shift (hypsochromic) (2–8 nm). It recommended that BSA and the test compounds in the ground state interact statically. In this case, the evaluated hyperchromism was found in the range of 47.79 to 64.70%. Furthermore, the findings obviously prove that conformational alterations may happen owing to non-covalent interactions such as H-bonds and electrostatic binding between substances and bovine serum albumin. The Benesi-Hildebrand Appendix A is also supported to evaluate the K_b_ values (Appendix A) [114]. The K_app_ findings for all substances were evaluated from the linear regression plot of [(A_∞_ − A_0_)/(A_x_ − A_0_)] vs. {1/[compound]} M^−1^ (Appendix A). The evaluated K_b_ findings for all test substances were in the subsequent order (Table 7): (**1b**) > (**2b**) > (**HL**) with ${\Delta G}_{b}^{{}^{\circ}}$ values from −22.3387 to −25.9330 kJmol^−1^. Complex (**1a**) is also clearly shown to have the greatest spontaneous binding efficacy with bovine serum albumin.

### 3.4. DFT and Molecular Modelling Properties

Ligand (**HL**) and (**1b**–**2b**) were optimized and confirmed to be real minima on the potential energy surface. In order to incorporate the solvent effect on the electronic structure of these compounds, the PCM solvation model was considered, which provides a reasonable compromise between accuracy and computing cost. PCM is based on the hypothesis that the solvent can be regarded as a continuous isotropic dielectric, in which a cavity is created to accommodate the solute. The solvation cavity is shaped by the solvent accessible surface of the solute molecule [115]. Figure 8 displays the optimized structures of the titled compounds. This figure clearly confirms the distorted octahedral geometry of complexes (**1b**–**2b**), which is consistent with the MEP maps of the individual subunits. It is worth noting that FMOs, i.e., the HOMO and LUMO, are widely relied on to probe into the reactivity of molecular systems. In fact, it has been extensively demonstrated that kinetic stability, which inversely relates to chemical reactivity, correlates with the energetic gap between the HOMO and LUMO. A rule of thumb is that the smaller the gap, the more reactive the compound is, and vice versa. Figure 9 displays the FMOs of the test compounds also provided in the same figure are the HOMO-LUMO gaps estimated in the gas phase and methanol using the B3LYP functional. HOMO-LUMO gaps range between 1.89 and 3.99 eV in the gas phase, and between 2.12 and 3.88 eV in methanol solution. Regardless of the environment, the same order of chemical reactivity is maintained, namely (**HL**) < (**2b**) < (**1b**), which indicates that complexes are more reactive than free ligand. In general, the implicit presence of solvent (methanol) molecules is reflected by the stabilization of both the HOMO and LUMO of the titled compounds, although to different extents. As a consequence, incorporating solvent effects slightly enhances the chemical reactivity of the ligand (**HL**) (HOMO-LUMO gap going from 3.99 to 3.88 eV), while decreasing that of complexes (**1b**–**2b**) (HOMO-LUMO gaps transiting in the range of 1.88–2.61 and 2.12–2.71 eV for complexes (**1b**) and (2**b**), respectively.

Although popular, the hybrid B3LYP functional, which integrates the energies of Lee-Yang-Parr correlation with Becke’s exchange, is blamed for both ignoring dispersion effects and the poor distinctive description of short and long-range interactions in the estimation of the electronic energy. Some other DFT functionals are known to overcome these difficulties, such as the range-separated CAM-B3LYP and the meta-hybrid M06-2X functionals. Note that the latter includes dispersion corrections thanks to empirical Grimme’s energy. Therefore, besides B3LYP, CAM-B3LYP and M06-2X functionals were considered to investigate the electronic structure and reactivity of the titled compounds. The results obtained are provided in the Appendix A. HOMO-LUMO gaps calculated using the CAM-B3LYP (M06-2X) functional are comprised of between 4.38 and 6.64 kcal/mol (4.06 and 6.90 kcal/mol) in the gas phase, and between 5.19 and 6.84 kcal/mol (4.75 and 7.36) in methanol. Note that B3LYP HOMO-LUMO gaps are slightly underestimated, while CAM-B3LYP and M06-2X lead to quantitatively comparable gaps. However, despite these discrepancies, all the DFT functionals predict the same order of chemical reactivity in both environments. Furthermore, docking studies of ligand (**HL**) and complexes (**1b**–**2b**) were performed on BSA, CT-DNA, and Mpro. Figure 10 displays schematic representations of the three biomacromolecules. The best docked guest-host complexes are shown in Figure 11. To begin with, BSA is a good model for the (human) serum albumin, a globular protein that contributes to the natural transportation of drugs and whose role is facilitated by its interaction with various cellular receptors. The docking of our compounds on BSA revealed binding energies in the range of −7.5 to −10.5 kcal/mol, which suggests spontaneous and favourable retention of these candidates in the active site of the protein. This retention was ensured by a collection of noncovalent interactions, comprising hydrogen bonds, π-sigma, π-π stacked and hydrophobic interactions. Hydrogen bonds were differentiated into conventional and non-conventional C-H…Y contacts (where Y is an electronegative centre). For instance, regarding the HL…BSA guest-host complex, conventional H bonds involved the Phe506 and Asn404 amino acid residues, while the Lys524 was found to receive a non-conventional C-H…O=C bonding interaction. On the other hand, DNA binding is the most imperative step to realizing the mechanism of anticancer activity [116]. Therefore, the ligand (**HL**) and complexes (**1b**–**2b**) were docked on CT-DNA. The docking scores ranging between −8.1 and −8.9 kcal/mol are reliable with the experimentally observed anticancer activity of these compounds and result from their ability to sneak between the two helixes. Finally, Mpro is a crucial enzyme involved in the replication of the SARS-CoV-2 virus. It is believed that inhibiting the activity of this biomolecule should mitigate the spread of the virus and thus constitutes one of the possible ways to tackle the ongoing CoVID-19 pandemic. However, for a compound to be able to block the action of Mpro, it has to favourably and strongly bind to its active site. Since the CoVID-19 outbreak, several strategies have been proposed to come up with promising candidates capable of inhibiting the activity of Mpro. Two of the most popular are the drug repurposing of FDA-validated antivirals [117] and the virtual screening of online libraries [118]. While these two approaches rely on existing molecules, another promising avenue consists in synthesizing new molecules and testing their potential inhibitory activity against Mpro [119]. Following the same vein, we have performed docking calculations of our synthesized candidates against Mpro. The returned docking scores ranged from −6.7 to −8.8 kcal/mol. The order of binding affinity was as follows: −6.7 (**HL**) < −7.1 (**2b**) < −8.8 (**1b**), which globally corroborates with the improved chemical reactivity of the complexes (**1b**–**2b**), while also suggesting that these complexes would constitute better inhibitors of Mpro as compared to free ligand (**HL**). The stability of guest-host complexes is guaranteed by several noncovalent interactions, namely hydrogen bonds and hydrophobic contacts. Important amino acid residues engaged in the interaction with the ligands include, without being restricted to, Glu166 (**2b**, **HL**), Gly143 (**HL**), Cyst45 (**1b**, **2b**, **HL**), His41 (**1b**) and Asn142 (**2b**), which are known to be located in the catalytic dyad of each Mpro protomer [120].

### 3.5. Assessment of Antioxidant Properties Using UV–Visible Spectral Titration

It is defined as the ability of any substance to put off or reduce the oxidation of substrate (proteins/lipids/DNA/carbohydrates of living cells) or free radical formation. The biological systems are shielded from the potential adverse effects of excessive oxidation by the oxidizable substrate. As a result, the free radical’s energy may be reduced, radical formation suppressed, or the chain propagation of lipid oxidation may be stopped in the initial stages. They also donate hydrogen or electrons to the free radicals, turning them into nontoxic or H_2_O molecules [121]. Free radicals are extremely unstable molecules with an unpaired electron that react quickly with other substances to collect nearby electrons to gain stability. This starts a chain reaction that eventually declines and leads to the loss of cellular function. In addition, antioxidants have the ability to scavenge these free radicals and stop the chain reaction, which prevents additional cellular damage [122,123,124,125]. The observed % of inhibition efficiency for all substances in term of IC_50_ findings for DPPH, hydroxyl radical, superoxide and nitric oxide power assays were revealed in Appendix A.

#### 3.5.1. Assessment of DPPH Radical Scavenging Property

The colour of an aqueous or methanol solution changes from purple to pale yellow when DPPH, a stable chromogen free-radical, combines with an antioxidant molecule and the donor groups’ hydrogen or electrons are swiftly taken up by DPPH. In this case, for the baseline correction, a blank DPPH solution in the absence of a test compound was employed, and 517 nm (ε = 8320 M^−1^ cm^−1^) was monitored to have a significant absorption maximum. It was found that when test compound concentrations (40–240 µM) increase, the DPPH radical inhibition increases as well. The DPPH^᛫^ radicals are reduced by an antioxidant compound (AH), in which the reduction of electronic absorbance for each and every compound was carefully noted at 517 nm [126]. The capacity to block radicals improves as the sample concentration increases. The percentage of maximum inhibition for all substances was found at 240 µM in the subsequent order: (**Ascorbic acid** (**AC**) (85.65) > (**1b**) 70.12 > (**2b**) 60.66 > (**HL**) 52.45. The IC_50_ values of standard ascorbic acid, complex (**1b**) and complex (**2b**) was found at 80 µM, 160 µM and 200 µM, respectively (Appendix A). Moreover, the % of scavenging for all substances is estimated with the supporting of Appendix A.

#### 3.5.2. Evaluation of Hydroxyl Radical Inhibition

H_2_O_2_ receives electrons from antioxidant molecules, which then neutralize it into a water molecule. OH^●^ inhibition capability was determined from the % of inhibition for all test substances at 230 nm. The % of inhibition for all samples at 240 µM was observed in the subsequent order: (**AC**) 78.83 > (**1b**) 62.05 > (**2b**) 57.83 > (**HL**) 50.68. The IC_50_ findings for standard ascorbic acid and complex (**1b**) observed at 160 µM and 200 µM, respectively. Complex (**1b**) showed the best antioxidant potency among others (Appendix A).

#### 3.5.3. Superoxide Scavenging Assay

A vital enzyme catalyst in the body’s defence against free radicals, superoxide dismutase (SOD) quickly and efficiently reduces toxicity and cellular damage by exchanging superoxide into water (or) harmless molecules. In many biological systems, the superoxide anion radical is a highly reactive and harmful species. One of the enzymes, Cu, Zn-superoxide dismutase, is in charge of this reactive superoxide. Numerous copper and other low molecular weight complexes have also been reported to have SOD-mimicking properties [127]. The % of inhibition for all substances was analysed at 590 nm. The obtained results are found in the following order: (**AC**) 84.85 > (**1b**) 68.21 > (**2b**) 60.02 > (**HL**) 50.42. However, complex (**1b**) revealed the best antioxidant potency among others, and standard ascorbic acid’s assessed IC_50_ findings were found to be 120 µM and complex (**1b**)’s values to be 200 µM, respectively (Appendix A).

#### 3.5.4. Assessment of Nitric Oxide Inhibition

The diffusible nitric oxide free radical is a crucial chemical mediator that assists in overcoming diverse chronic human diseases. The NO^᛫^ free radical scavenging potential for all test samples was also carried out at 546 nm. When the test sample concentration rises, the nitric oxide inhibition effectiveness also increases. The measured % of nitric oxide radical scavenging capability for all samples at 240 µM was obtained in the subsequent order, (**AC**) 72.73 > (**1b**) 66.21 > (**2b**) 59.88 > (**HL**) 51.62. Complex (**1b**) showed superior antioxidant efficacy among the complexes, and standard ascorbic acid’s IC_50_ values were observed to be 160 M and 200 M, respectively (Appendix A).

### 3.6. Evaluation of Antimicrobial Properties

The current research has a curious focus on the in vitro antimicrobial properties of biological systems because these studies are crucial for developing effective antibacterial and antifungal medications. The obtained clear inhibition zone (mm) values for various bacterial and fungal species in the samples were revealed in Figure 12, and the evaluated findings were enclosed in Table 8. The obtained findings propose that all metal complexes (**1b**–**2b**) demonstrate significantly greater antimicrobial properties than free ligand (**HL**) against the chosen microorganism, and they are contrasted with standard drugs such as amikacin and streptomycin for bacteria, and ketokonazole and amphotericin *B* for fungal species. Therefore, the altered structure and reactivity of the ligand upon coordination with the metal centre enhances their antibacterial capabilities [128,129]. In addition, the azomethine group of the compounds may form an H-bond with the active centres of cell constituents, inhibiting normal cell function as part of their mode of action. Based on the chelation theory proposed by Overtone and Tweedy, it can also be elucidated [130]. Chelation theory states that the partial exchanging of the positive charge of the metal centre with donor clusters and overlap of the ligand orbitals will reduce the greater degree of the metal ion’s polarity, which ultimately leads to the delocalization of π and d electrons over the entire chelated ring system. By raising the size of the metal ion due to the retarding of polarization, chelation may also escalate the complexes’ lipophilic characteristics, which further stimulate the lipid membrane permeability and break down the bacteria’ enzymes responsible for cell wall formation, therefore slowing down the regular cell processes. By preventing the production of cell walls/proteins/DNA, including by obstructing folate metabolism and the cytoplasmic membrane, antimicrobial drugs frequently either fully eliminate microbes or only prevent their cell growth. Additionally, the samples’ mode of action may be employed in disrupting the cell’s respiration process by forming an H-bond in the course of the morpholine-fused iminic group coordinated with the active metal centre of its components, inhibiting proliferation of the cell. The presence of co-ligands, the nature of the metal ion, the nature of the ligand, coordinating sites, concentration, hydrophilicity, and lipophilicity, as well as other significant factors, have a significant impact on the antibacterial activity of the compounds, even though chelation predominately affects their biological behaviour. It is also revealed that the enhanced antibacterial activity could be attributed to changes in pharmacological kinetics, conductivity, steric, electronic, solubility, metal-ligand bond length, etc. The effectiveness of an antibacterial agent is also significantly influenced by these variables. However, it may be concluded that the antibacterial activity of metal complexes is not just due to chelation but rather a complicated combination of numerous factors [131]. The difference in the antimicrobial efficacy of some of the compounds towards various microorganisms depends on the impermeability of the cells of the germs or the diversity of ribosomes in the microbes [132]. The % of inhibition of all substances is estimated from Appendix A.

### 3.7. Assessment of Cytotoxic Properties

The cytotoxic efficacy of medications is mostly based on their capacity to cause either apoptosis or necrosis. Successful anticancer medications typically activate a number of cellular processes to eliminate cancerous cells. The therapeutic qualities of the medicine can also be impacted by other pharmacodynamic parameters such as solubility and interactions with other biomolecules [133,134,135]. Moreover, cellular viability or metabolic properties can be measured using the MTT assay, which is a powerful and consistent method for cytotoxic properties. The cytotoxic efficacy of all test compounds was evaluated by the MTT assay against A549, HepG2, MCF-7, and NHDF cell lines [136]. As per the colorimetric approach, the IC_50_ findings of all compounds were evaluated according to the % of cell viability or growth inhibition [137]. Even though the complexes demonstrate higher activity compared to free ligand against some cancer cell lines, the NHDF cell line is only mildly perturbed by cisplatin. Nevertheless, complex (**1b**) was exposed to the highest cytotoxic potential compared to others [138]. The acquired findings were in the subsequent sequence (Table 9): (**Cisplatin**) > (**1b**) > (**2b**) > (**HL**) (Figure 13 and Table 9). The cytotoxic effectiveness is dependent upon the DNA binding modalities, the structure-activity relationship, as well as the test drug concentrations and incubation period exposure [139]. In addition, it is suggested that these complexes consist of the morpholine-fused primary aromatic and secondary 1,10-phenanthroline planar systems connected with a metal centre, which facilitates their simple insert within the base pairs of DNA. As per Tweedy’s chelation theory, coordination between the ligands and metal ions results in charge equilibration, which diminishes the polarity of the metal ions and further causes the capability of the test complexes to pass via the cell membrane lipid layer. Thus, it put off the synthesis of cell-wall/protein/nucleic acid [140]. The measured % of the inhibition results of growth for these compounds is summarized in Table 9. Additionally, the results of DNA binding tests using these complexes, including gel electrophoresis, UV–Visible Spectral titration, hydrodynamic, emission, and CV findings, were in good agreement with the findings of cytotoxicity. The Equations (S39) and (S40) were supported to measure the % of growth inhibition and the % of cell viability, respectively (Appendix A). The complex (**1b**) has been proven to have greater biological efficiency among others based on the Lewis acid character, solubility, conductivity, bond length of metal-ligand, charge, electron density, dipole moment, intermolecular H-bonding, and proton transfer equilibrium, etc. These significant elements might also contribute to the increased biological activity. The Cu^+^ ion is unique among the transition metals owing to its size, d^10^ electronic configuration, softness, polarizability, and flexible characteristics of the distorted coordination geometry. The effect of the Cu^+^ is a reduced form of the complexes’ structure, symmetry, and function that results in increased biological efficacy. Depending on the ligand donor selected, Cu^+^ can also be reduced as an intervening molecule. Despite of having the same electronic configuration for both Cu^+^ and Zn^2+^ ions, Cu^+^ is softer and more flexible than Zn^2+^ ions. Hence, the copper complex may promote DNA damage as well as inhibiting DNA repair, which also exposed the double-edged effects. According to numerous research findings, via cell apoptosis or enzyme inhibition, copper complexes have been proven the great cytotoxic efficacy.

## 4. Conclusions

All synthesized compounds are treated with diverse analytical, spectral, and X-ray diffraction analyses. The observed results of the complexes (**1b**–**2b**) proposed an octahedral geometry. The gel electrophoresis results showed that complex (**1b**) revealed excellent metallo nuclease efficacy under the H_2_O_2_ environment. The observed DNA binding properties of all compounds by spectro-electro-hydrodynamic and fluorometric titrations disclose that complexes (**1b**–**2b**) could bind with DNA via intercalation. The observed BSA binding constants of all samples assign that the complexes could interact with BSA in static mode, which is further supported by FRET measurements. Complex (**1b**) also possessed the best DNA/BSA binding affinities compared to others. The electronic configuration data of these substances were attained from DFT and their molecular docking studies on the interacting affinity of these substances against DNA/BSA/SARS-CoV-2. The findings revealed that the metal complexes bind spontaneously inside the active sites of these biomolecules. In addition, the enhanced reactivity of the metal complexes with respect to the ligand (**HL**) is well accounted for in the context of the FMO theory. The theoretical measurements for all substances were reported to be in excellent accord with the experimental findings. The antimicrobial properties demonstrated that the metal complexes have highly significant inhibition potency than free ligand (**HL**). The scavenging properties put forward by complexes stood out as having greater potential to scavenge radicals than free ligand. The observed in vitro anti-cancer characteristics findings for all substances and cisplatin (**CP**) revealed that complex (**1b**) revealed the best cytotoxic efficiency among others, and the damaged normal cell was found to be less compared to the standard drug cisplatin. Complex (**1b**) might function as a brand-new class of anticancer agent in the future.

## Figures and Tables

**Figure 1 biomolecules-12-01883-f001:**
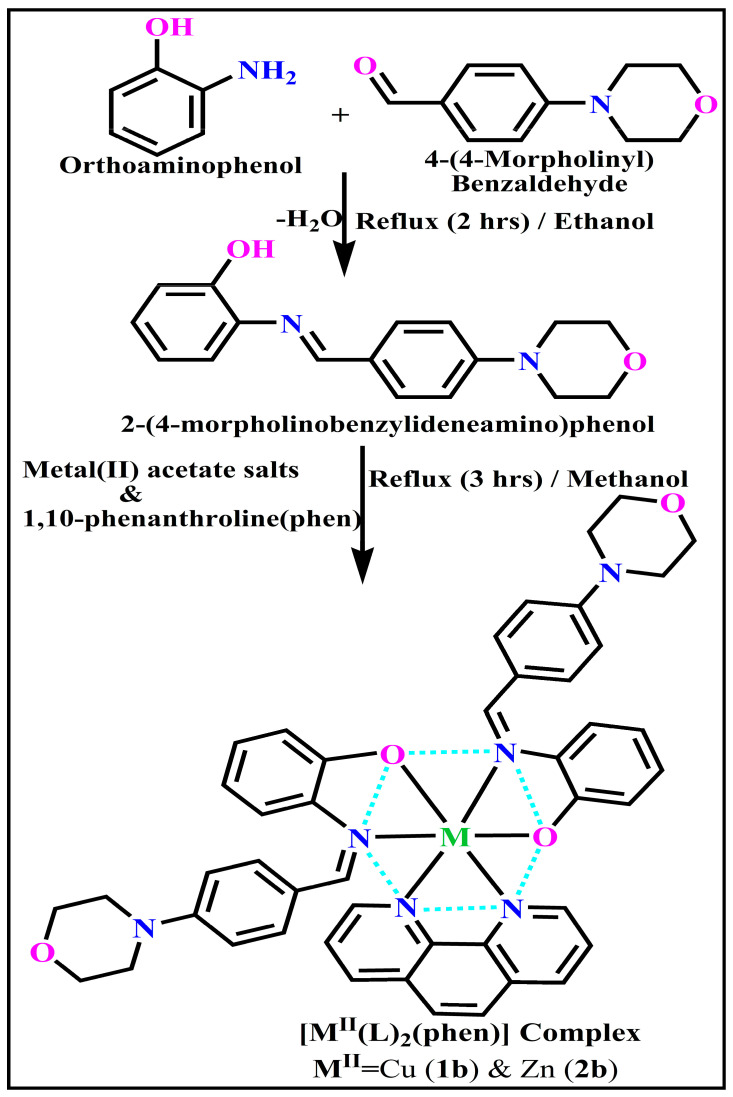
The proposed structure of complexes (**1b**–**2b**) **[M^II^(L)_2_ (phen)]**.

**Figure 2 biomolecules-12-01883-f002:**
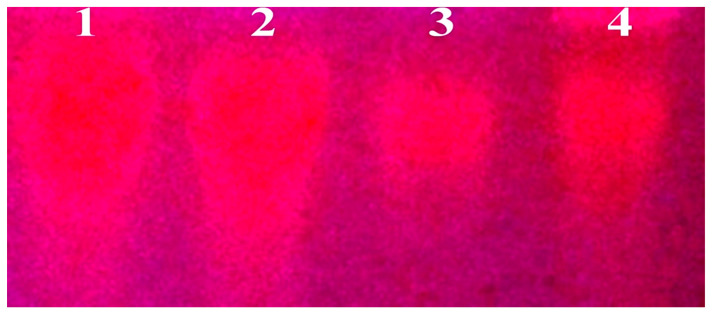
The ethidium bromide displacement assay: gel electrophoresis showing the chemical nuclease activity of CT-DNA by the synthesized ligand (**HL**) and its complexes (**1b**–**2b**) in the presence of hydrogen peroxide. Lane: 1 DNA alone + H_2_O_2_; Lane: 2 ligand (**HL**) + DNA + H_2_O_2_; Lane: 3 complex (**1b**) + DNA + H_2_O_2_; Lane: 4 complex (**2b**) + DNA + H_2_O_2_.

**Figure 3 biomolecules-12-01883-f003:**
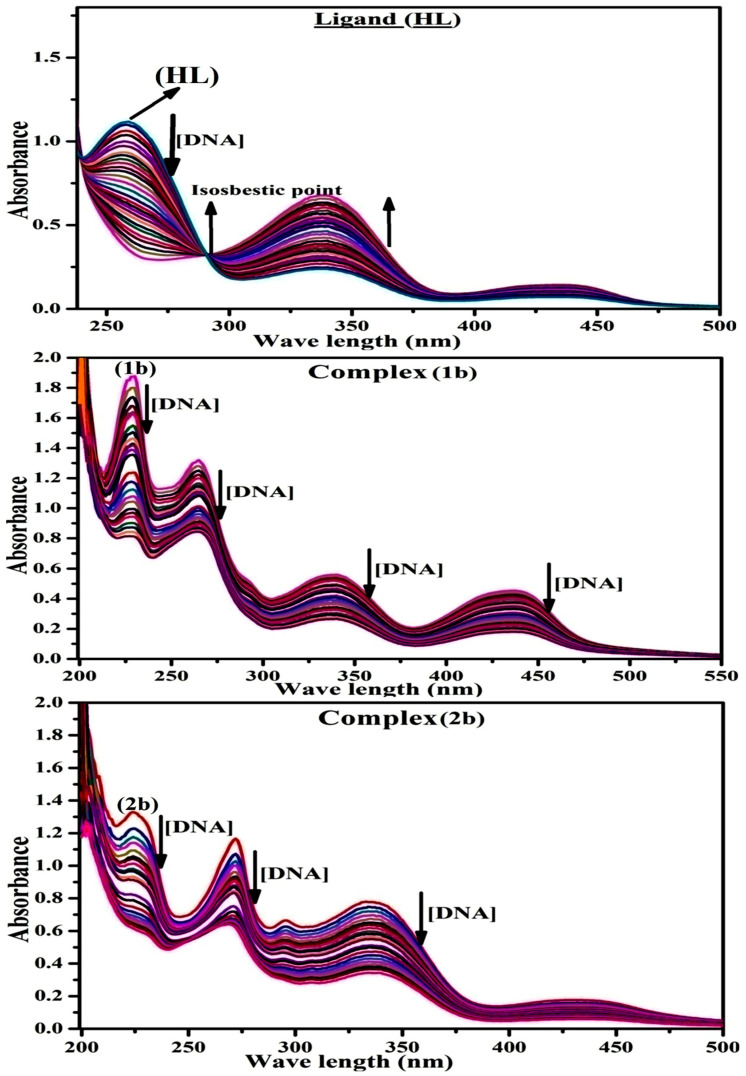
The absorption spectra of ligand (**HL**) and its complexes (**1b**–**2b**) in Tris-HCl buffer pH = 7.2 at 25 °C in the presence of increasing amount of CT-DNA. The arrow indicates the changes in absorbance upon increasing the CT-DNA concentration and another arrow shows isosbestic points indicates that equilibrium is established between DNA and complexes.

**Figure 4 biomolecules-12-01883-f004:**
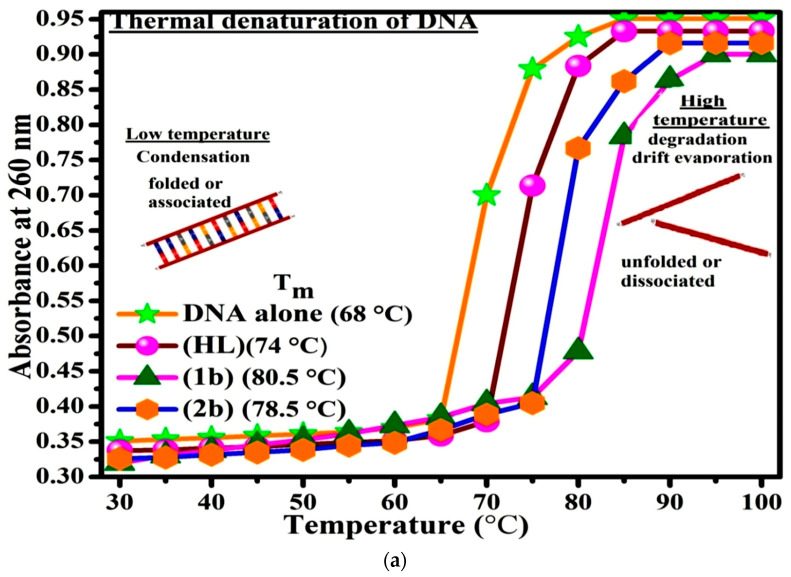
(**a**) DNA thermal denaturation profile at 260 nm in the absence and presence of ligand (**HL**) and complexes (**1b**–**2b**) in 5 mM Tris-HCl/50 mM NaCl buffer pH = 7.2, [DNA]/[Complex] = 1(R), the denaturation temperature (T_m_) was taken as the mid-point of the hyperchromic transition. (**b**) The derivative melting curve for DNA thermal denaturation at 260 nm in the absence and presence of compounds in 5 mM Tris-HCl/50 mM NaCl buffer pH = 7.2, [DNA]/[Complex] = 1(R).

**Figure 5 biomolecules-12-01883-f005:**
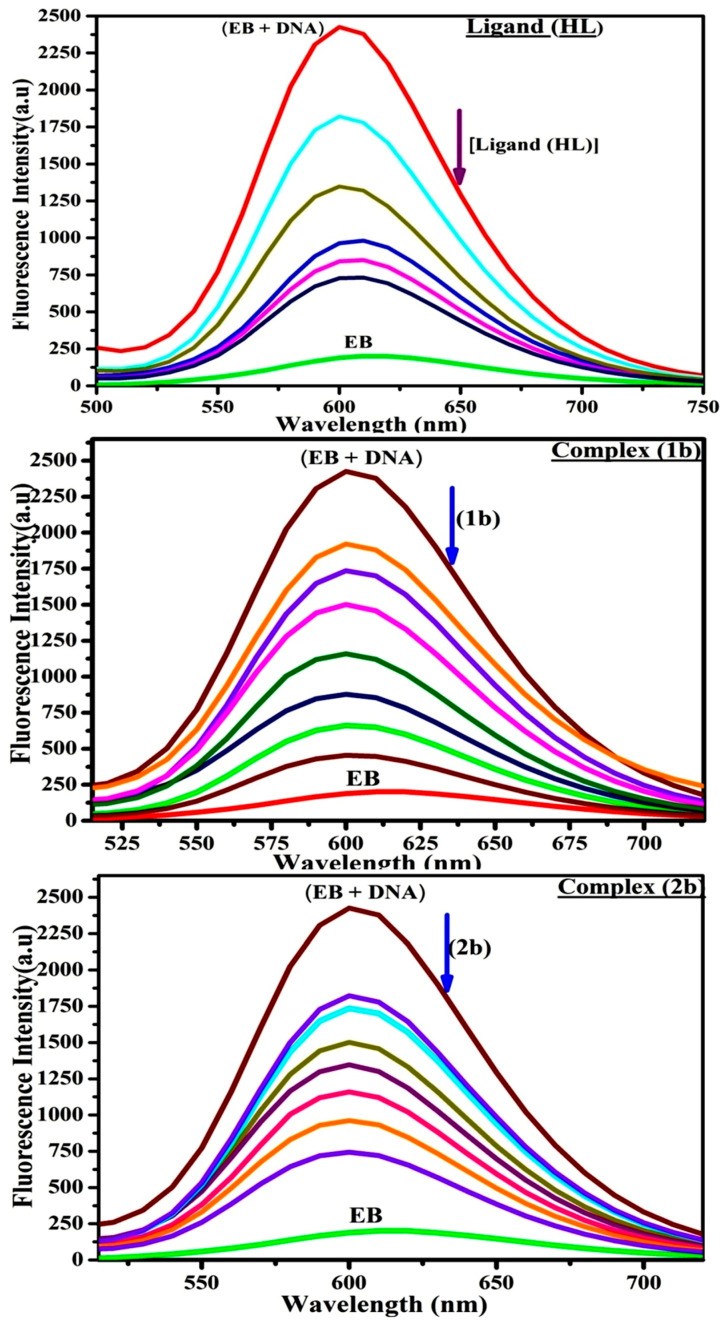
The fluorescence quenching curve of **EB** bound DNA in the presence of ligand (**HL**) and complexes (**1b**–**2b**). Concentration [DNA] = 240 μM, and [compound] = 0–240 μM. The fluorescence emission spectra of the EB-DNA adduct absorbed at 610 nm.

**Figure 6 biomolecules-12-01883-f006:**
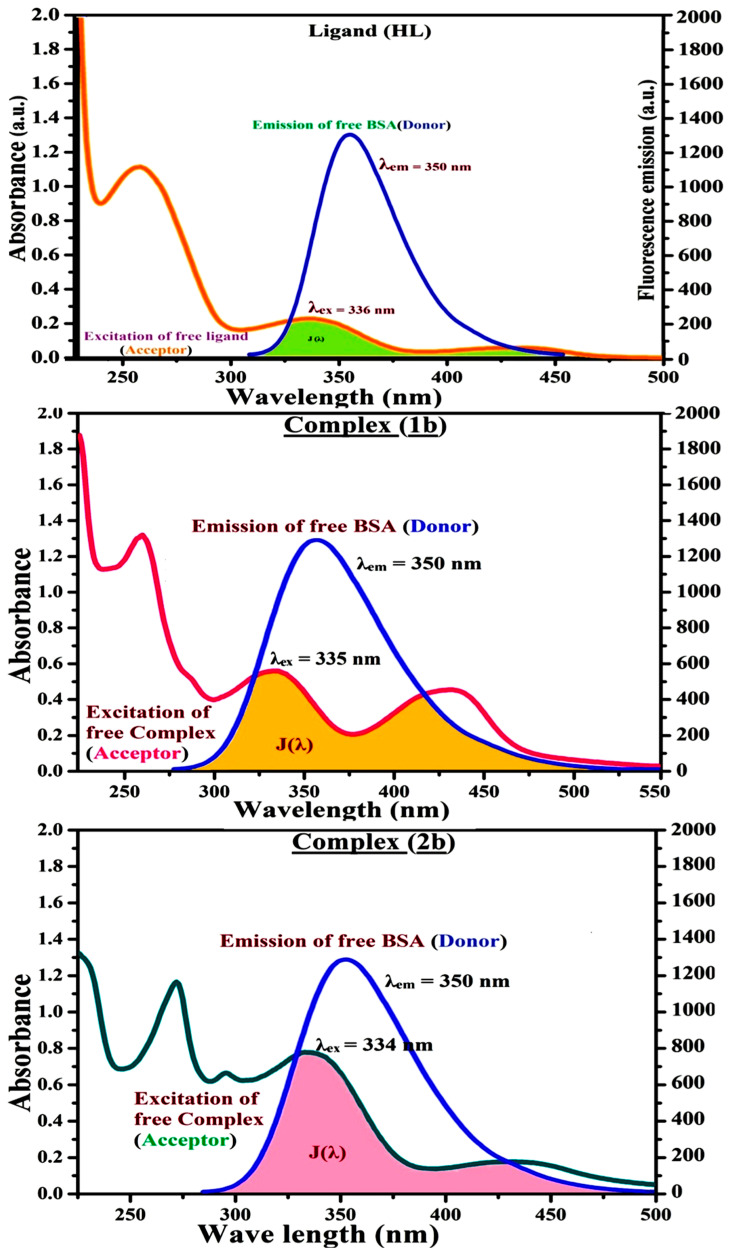
The overlap of UV–Vis spectra of ligand (**HL**) and its complexes (**1b**–**2b**) (**Acceptor**) at 334–336 nm with fluorescence emission spectrum of BSA (**Donor**) at 350 nm.

**Figure 7 biomolecules-12-01883-f007:**
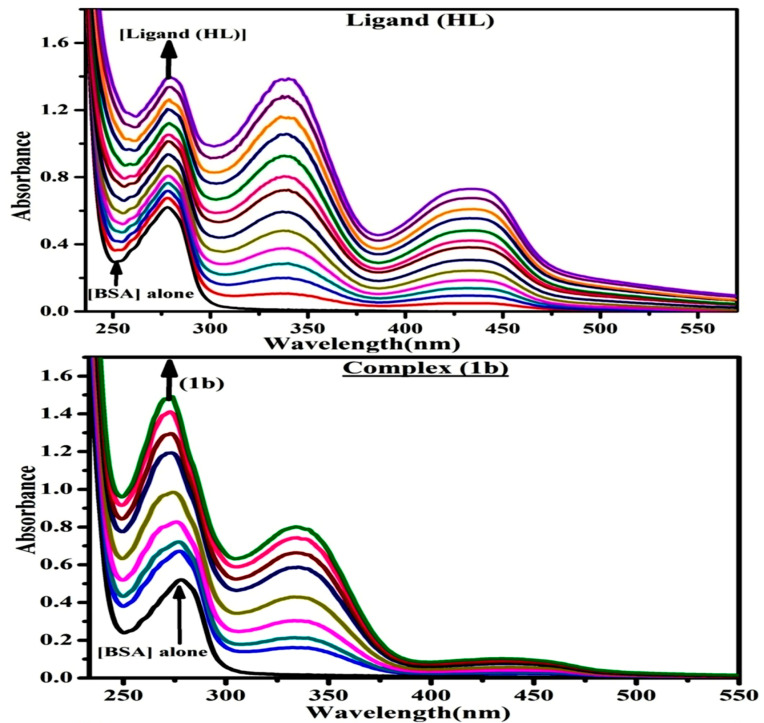
The absorption spectra of BSA alone and in the presence of increasing amount of ligand (**HL**) and complexes (**1b**–**2b**) in Tris-HCl buffer pH = 7.2 at 25 °C. The arrow indicates the changes in absorbance upon increasing the compound concentration.

**Figure 8 biomolecules-12-01883-f008:**
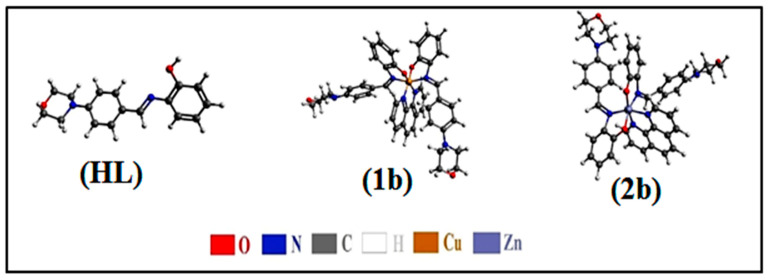
The optimized geometries of the free ligand (**HL**) and complexes (**1b**–**2b**).

**Figure 9 biomolecules-12-01883-f009:**
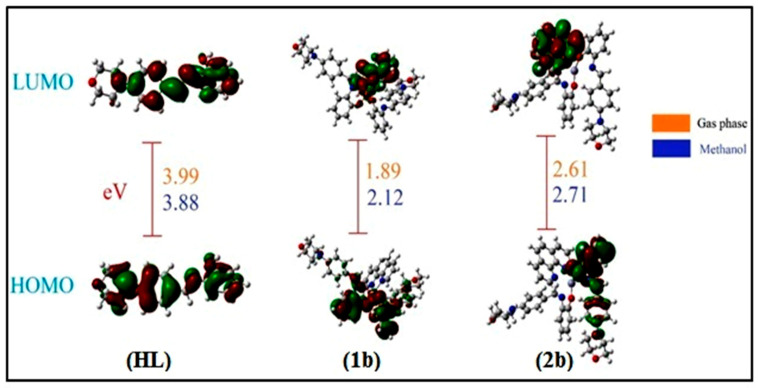
The frontier molecular orbitals of the free ligand (**HL**) and complexes (**1b**–**2b**). HOMO-LUMO gaps in the gas phase and methanol are provided.

**Figure 10 biomolecules-12-01883-f010:**
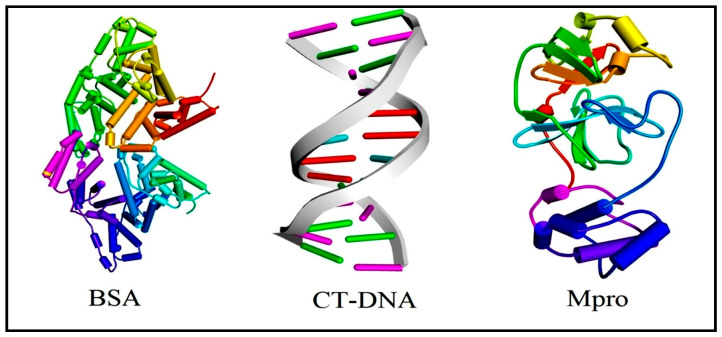
The schematic structures of the three biomacromolecules considered for docking studies of ligand (**HL**) and complexes (**1b**–**2b**). Only one protomer of the dimeric Mpro enzyme is provided.

**Figure 11 biomolecules-12-01883-f011:**
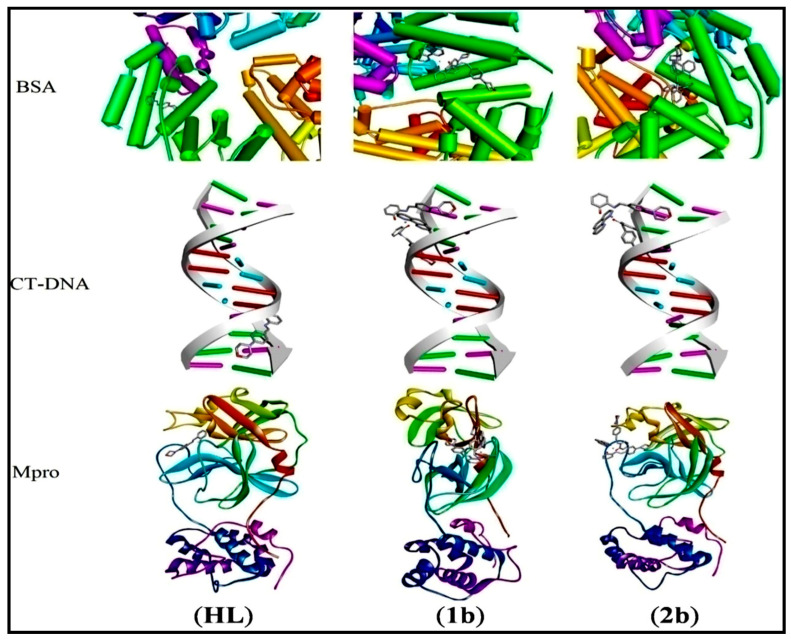
The best docking poses of the ligand (**HL**) and its complexes (**1b**–**2b**) against BSA, CT-DNA and Mpro.

**Figure 12 biomolecules-12-01883-f012:**
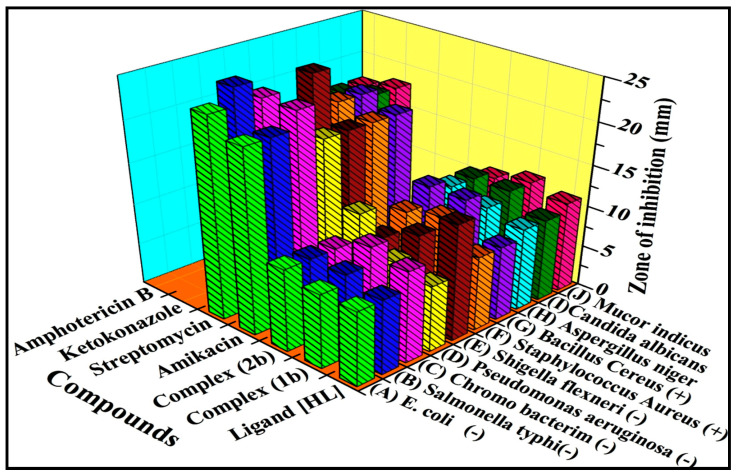
Histogram showing the comparative antimicrobial activities of ligand (**HL**) and its mixed ligand complexes (**1b**–**2b**) by Agar disc diffusion method.

**Figure 13 biomolecules-12-01883-f013:**
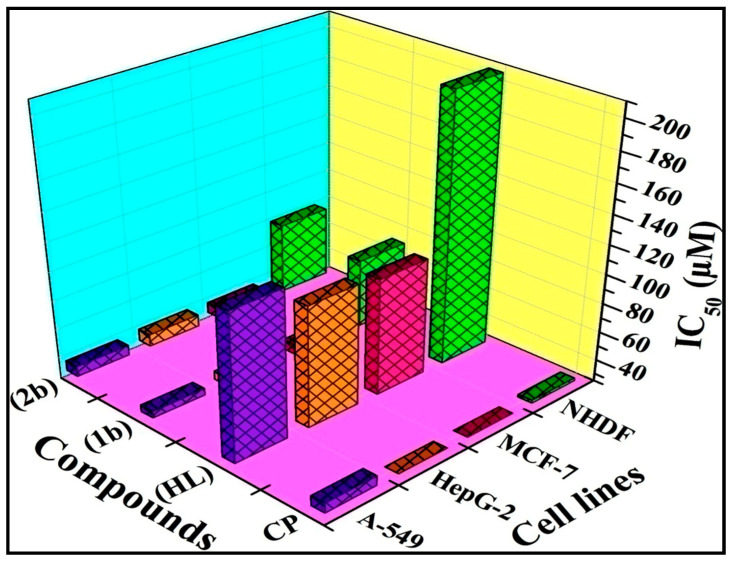
The comparison of cytotoxic effects of ligand (**HL**) and its complexes (**1b**–**2b**) with standard drug cisplatin (**CP**) against cancer and normal cell lines. Error limits ± 2.5–5.0% (*p* ≤ 0.05).

**Table 1 biomolecules-12-01883-t001:** The absorption spectral parameters of ligand (**HL**) and its complexes (**1b**–**2b**) bound to CT-DNA.

Compounds	λ_max_ (nm)	Δλ nm (% H)	K_b_ × 10^4^ M^−1^			${\Delta G}_{b}^{{}^{\circ}}$ (kJmol^−1^)		
	Free (Bound)		WS–I (WS–II)	BH–I (BH–II)	SK–I (SK–II)	WS–I (WS–II)	BH–I (BH–II)	SK–I (SK–II)
(**HL**)	336 (340)	4 (46.88)	1.5169 (1.5480)	1.0545 (1.0545)	0.8775 (1.5618)	−23.85 (−23.90)	−22.95 (−22.95)	−22.50 (−23.92)
(**1b**)	335 (342)	7 (56.88)	2.0610 (2.1357)	2.5359 (2.5251)	2.0314 (2.1747)	−24.61 (−24.70)	−25.12 (−25.11)	−24.58 (−24.74)
(**2b**)	334 (339)	5 (55.75)	1.9930 (2.0811)	2.0131 (2.0035)	1.9471 (2.0897)	−24.53 (−24.64)	−24.55 (−24.54)	−24.47 (−24.65)

Hypochromism $\left(\%\;H\right)=\frac{\left({ɛ}_{b}-{ɛ}_{f}\right)}{{ɛ}_{f}}\times 100$; ɛ_f_ → extinction coefficient of free complex [ɛ_f_ = (A/C), where, A → absorption of free compounds, C → concentration of free compounds and *ɛ_b_* → extinction coefficient of the fully bound to DNA, WS → Wolfe-shimmer, BH → Benesi–Hildebrand methods (BH–I and II), SK → Sakthi–Krause methods (SK–I and II), Gibbs free energy Change ${\Delta G}_{b}^{{}^{\circ}}=-{\mathrm{RT}\;\ln\;K}_{b}$, K_b_ = Intrinsic DNA binding constant determined from the UV–Vis absorption spectral titration, R → Universal gas constant = 1.987 cal K^−1^mol^−1^ (or) 8.314 J K^−1^ mol^−1^, T = 298 K, Error limit ± 2.5%.

**Table 2 biomolecules-12-01883-t002:** The thermodynamic profiles of electronic absorption spectra for the binding of ligand (**HL**) and its complexes (**1b**–**2b**) to CT–DNA.

Compounds	T_m_ °C (K)	ΔT_m_ °C	Binding Constant K_r_ (or) K_1_ @ 298 K (M^−1^)	Binding Constant K_m_ (or) K_2_ @ T_m_ K (M^−1^)	ΔH° (kcal mol^−1^)	ΔS° (cal mol^−1^)	ΔG° (kcal mol^−1^)
(**HL**)	74 (347.0)	6	1.5169 × 10^4^	1.4625 × 10^3^	−9.8083	−13.7851	−5.0249
(**1b**)	80.5 (353.5)	12.5	2.0610 × 10^4^	2.3722 × 10^3^	−8.1538	−7.6237	−5.4588
(**2b**)	78.5 (351.5)	10.5	1.9930 × 10^4^	2.1862 × 10^3^	−8.5979	−9.1806	−5.3709

CT DNA melting temperature (T_m_) = 68 °C (341 K); ΔT_m_ → Melting temperature changes between complexes and DNA alone in the thermal denaturation experiments; ln$\left[\frac{K_{2}}{K_{1}}\right]=-\frac{\Delta H{}^{\circ}\;}{R}\;\left[\frac{1}{T_{2}}-\frac{1}{T_{1}}\right]\;$→ Enthalpy Change (ΔH°) = R ·$\;\left[\frac{T_{m}{\;T}_{r}}{T_{m}-{\;T}_{r}}\right]\cdot \;$ln$\left[\frac{K_{m}}{K_{r}}\right]$, where K_1_, T_1_ and K_2_, T_2_ are denoted as K_r_ (binding constant), T_r_ (DNA Melting temperature) @ 298 K and K_m_, Tm @ T_m_, respectively, in the DNA-compound system, R → Universal gas constant = 1.987 cal K^−1^mol^−1^; Gibb’s free energy (ΔG°) = −R · T_m_ · ln K_m_; Entropy change (ΔS°) = $\left[\frac{\Delta H{}^{\circ}-\Delta G{}^{\circ}\;}{T_{m}}\right]$.

**Table 3 biomolecules-12-01883-t003:** The relative specific viscosity vs. [Compound]/[DNA].

Compounds	Binding Ratio (R) = [Complex]/[DNA]						
	0.2	0.4	0.6	0.8	1.0		
	Relative Specific Viscosity (*η*/*η*_0_)^1/3^					Slope	R^2^
**EB** (Control)	1.01	1.35	1.63	1.82	1.99	1.2859	0.97379
(**HL**)	0.61	0.67	0.75	0.85	1.01	0.4616	0.95002
(**1b**)	0.82	1.10	1.25	1.45	1.71	1.0805	0.98652
(**2b**)	0.80	0.91	1.16	1.21	1.53	0.8400	0.94373

*η* → Specific viscosity of DNA in the presence complex, *η*_0_ → Specific viscosity of DNA alone, Error limit ± 2.0%.

**Table 4 biomolecules-12-01883-t004:** The binding constants and binding sites for the interaction of ligand (**HL**) and mixed ligand complexes (**1b**–**2b**) with EB–DNA at pH = 7.4 from ethidium bromide displacement assay by fluorescence spectral titration.

Compounds	Binding Constants for DNA/BSA with Test Compounds									
	Stern–Volmer Methods for DNA Binding Properties (Stern–Volmer Methods for BSA Binding Properties)						LWB Method K_LB_ × 10^4^ M^−1^	Scatchard Analysis		K_app_ × 10^7^ M^−1^
	Method–I		Method–II							
	K_q_ × 10^12^ M^−1^s^−1^	K_SV_ × 10^4^ M^−1^	K_ass_ × 10^4^ M^−1^	n	${\Delta G}_{b}^{{}^{\circ}}$ (kJM^−1^)	P		K_SA_ × 10^4^ M^−1^	n	
(**HL**)	1.1636 (2.639)	1.1636 (2.639)	0.9882 (1.524)	0.976 (0.958)	–22.790 (–23.86)	0.090 (0.225)	0.6985	2.3194	1.096	0.4200
(**1b**)	4.0303 (9.564)	4.0303 (9.564)	3.1817 (7.199)	1.088 (1.021)	–26.942 (–27.71)	0.128 (0.322)	2.1654	3.3302	1.008	1.0000
(**2b**)	2.0669 (7.935)	2.0669 (7.935)	1.9688 (5.984)	1.083 (1.009)	–24.873 (–27.25)	0.124 (0.285)	1.8218	3.2941	0.909	0.4804

K_SV_ → Stern-Volmer binding constant; K_ass_ → Association binding constant; K_app_ → Apparent binding constant, $K_{\mathrm{app}}=K_{\mathrm{EB}}\frac{\left[\mathrm{EB}\right]}{\left[\mathrm{compound}\right]}\;=\;\frac{500}{\left[\mathrm{compound}\right]}$, K_EB_ = 10^7^ M^−1^ at the concentration of 50 µM EB; Gibb’s free energy Change ${\Delta G}_{b}^{{}^{\circ}}=-{\mathrm{RT}\;\ln\;K}_{\mathrm{ass}}$; K_q_ → Bimolecular quenching rate constant or Stern–Volmer dynamic quenching rate constant $\left(K_{q}=\frac{K_{\mathrm{SV}}}{\tau_{0}}\right)$, average life time of biomolecular quenching in the absence of a quencher ($\tau_{0}$) = 10^−8^ S; Gibb’s free energy Change ${\Delta G}_{b}^{{}^{\circ}}=-{\mathrm{RT}\;\ln\;K}_{\mathrm{ass}}$ (Where, R = 8.3144 kJmol^−1^, T = 298 K); K_LB_ → Lineweaver-Burk (LWB) binding constant; K_SA_ → Scatchard Association binding constant; K_app_→ Apparent binding constant; n = → the number of binding sites; P → Ratio of fluorescence quantum efficiency of DNA bound and free complex $\left(P=\frac{\phi_{b}}{\phi_{f}}\right)$, which is obtained as intercept from plot F/F_0_ vs. 1/[DNA], Error limit ± 2.5%.

**Table 5 biomolecules-12-01883-t005:** Fluorescence Resonance Energy Transfer (FRET) parameters for Donor (BSA)–Acceptor (compound) systems.

Compounds	J × 10^−14^ (LM^−1^cm^3^)	R_0_ (nm)	E	r (nm)	k_ET_ (J/s)	B (M^−1^cm^−1^)
(**HL**)	0.8215	2.4400	0.3462	2.6685	5.8443	5339.79
(**1b**)	0.6852	2.3673	0.2769	2.7780	3.8294	4999.08
(**2b**)	0.9460	2.4980	0.3231	2.8257	4.7730	5650.81

J → Normalized spectral overlap integral between the emission spectrum of donor (BSA) and the absorption spectrum of acceptor (complex), R_0_ → Critical distance at which the efficiency of resonance energy transfer (50%) $R_{0}=\sqrt[{6}]{0.2569\times 10^{-25}}$ J, average refracted index of medium (n) = 1.36, fluorescence quantum yield of the donor (Φ) = 0.15, orientation factor related to geometry of the donor and acceptor of the dipoles (K^2^) = 2/3 for the complex-BSA interaction (or) relative factor of the distinctive orbital interactions based on orbital overlap between donor and acceptor, E → Efficiency of energy transfer, E=1−FF0, F and F_0_ are the fluorescence intensity of BSA in the presence and absence of complex, r → the donor-acceptor separation relative to their van der Waals radii L (nm), $r=\sqrt[{6}]{\left[(R_{0}^{6}/E)-R_{0}^{6}\right]}$, k_ET_ → Rate of exchange resonance energy transfer, B → Average brightness of the complex-BSA system, B = [(Φ_1_ε_1_+ Φ_2_ε_2_)/2], ε → molar absorption (or) extinction coefficient of the acceptor at λ, ε = 43,824 LM^−1^ cm^−1^ for donor (BSA) and ε values for acceptors = 27,373.20 (**HL**), 22,830.40 (**1b**) and 31,520.09 (**2b**). B value of free BSA = 6573.60 M^−1^ cm^−1^.

**Table 6 biomolecules-12-01883-t006:** The redox potential profiles for interaction of CT-DNA with complexes (**1b**–**2b**).

Compounds	ΔE_P_ (V)	E° (or) E_1/2_ (V)	$\frac{K_{\left[red\right]}}{K_{\left[oxi\right]}}$ Found (I) (Calcd)	$\frac{I_{pa}}{I_{pc}}$ Free (Bound)	D_o_ × 10^−5^ cm^2^ s^−1^	K_b_ × 10^4^ M^−1^ (Methods)			S (bp)
	Free (Bound)	Free (Bound)			Free (Bound)	I Red (Oxi)	II	III	
(**HL**)	0.7420 (0.8890)	0.3490 (0.3680)	1.1125 (2.0964)	1.4295 (1.3424)	2.8570 (2.5809)	0.5874 (0.528)	0.2477	0.4837	0.446
(**1b**)	0.2366 (0.2031)	0.7015 (0.7317)	1.2053 (3.2434)	0.8494 (0.7920)	4.4035 (4.0688)	2.0878 (1.732)	3.5514	2.1695	0.267
(**2b**)	0.4261 (0.4350)	0.5372 (0.5712)	1.1168 (3.7609)	0.6072 (0.6464)	4.0597 (3.6573)	1.6317 (1.461)	3.2242	1.8300	0.248

ΔE_P_ → Peak-to-peak separation = (E_Pa_ − E_Pc_), E° (or) E_1/2_ → Formal electrode potential = ½ (E_Pa_ + E_Pc_), $E_{s}^{{}^{\circ}}=\left(E_{b}^{{}^{\circ}}-E_{f}^{{}^{\circ}}\right)$
$E_{b}^{{}^{\circ}}$ and $E_{f}^{{}^{\circ}}$ are the formal electrode potential of the M(II)/M(I) couple in the free and bound forms, respectively. $E_{s}^{{}^{\circ}}$ = 0.019 V (**HL**), 0.0302 V (**1b**), 0.034 V (**2b**). $\left(\frac{K_{\left[\mathrm{red}\right]}}{K_{\left[\mathrm{oxi}\right]}}\right)=\mathrm{Ant}.\log\;n\left(\frac{E_{s}^{{}^{\circ}}}{0.0591}\right)$, where n = 1, D_0_ → Diffusion coefficient (cm^2^ s^−1^) of the M(I)/M(II) couple in the free and bound forms, respectively. I_pa_ →Anodic peak current, I_pc_ → Cathodic peak current. K_+_ → Binding constant of reduction process, K_2+_ → Binding constant of oxidation process, S → Binding site size of base pairs (bp) with a molecule of complex, Scan rate →100 mV s^−1^, Binding constant (K_b_) values observed from the linear plots of log (1/[DNA]) vs. log (I/I_0_ − I) for oxidation and reduction, (I_0_ − I_DNA_)/I_DNA_ = C_p_/C_f_ vs. [DNA] and Ip^2^ vs. (I_po_^2^ − I_p_^2^)/[DNA] by methods-I, II and III, respectively. Diffusion coefficient ($D_{0})=$
$\sqrt{7.51\times 10^{-5\;}(\mathrm{Slope})}$ (Where, n = 1, Charge transfer coefficient (α) ≈ 0.5 for quasi-reversible systems and calculated from Bard–Faulkner relation [α = 47.7/(E_Pa_ − E_P/2_)], C_o_ → Bulk concentration of the compound, A ≈ 0.07 cm^2^.

**Table 7 biomolecules-12-01883-t007:** The absorption spectral parameters of ligand (**HL**) and its complexes (**1b–2b**) bound to BSA.

Compounds	λ_max_ (nm)		Δλ (nm)	Chromism (% H)	Binding Constant K_app_ × 10^4^ M^−1^ by BH Method	${\Delta G}_{b}^{{}^{\circ}}$ (kJmol^−1^)
	Free	Bound				
(**HL**)	278	276	2	47.79	0.8237	−22.3387
(**1b**)	278	270	8	64.70	3.5140	−25.9330
(**2b**)	278	270	8	62.16	2.1878	−24.7590

Hyperchromism $H\;\%=\frac{\left(A_{\infty }-A_{0}\right)}{A_{\infty }}\times 100$; A_0_ → Absorbance of BSA alone at 278 nm, A_∞_ → Absorbance of the fully bound form of BSA with complex or ligand and A_x_ → Absorbance of BSA in the addition of different concentration of complex or ligand, Gibb’s free energy Change ${\Delta G}_{b}^{{}^{\circ}}=-{\mathrm{RT}\;\ln\;K}_{\mathrm{app}}$ (Where, R = 8.3144 KJmol^−1^, T = 298 K); K_app_ → Apparent binding constant determined from the UV–Vis absorption spectral titration, Error limit ± 2.5.

**Table 8 biomolecules-12-01883-t008:** The evaluation of Antimicrobial activities (diameter of clear zone inhibition in mm) (Inhibition %) of the ligand (**HL**) and its complexes (**1b**–**2b**).

Compounds	Antibacterial Activity							Antifungal Activity		
	A	B	C	D	E	F	G	H	I	J
Ligand (**HL**)	9 (33)	9 (33)	11 (45)	8 (25)	14 (57)	9 (33)	9 (33)	10 (40)	10 (40)	11 (45)
Complex (**1b**)	9 (33)	10 (40)	11 (45)	9 (33)	17 (65)	11 (45)	10 (40)	10 (40)	11 (45)	10 (40)
Complex (**2b**)	10 (40)	10 (40)	11 (45)	08 (25)	14 (57)	11 (45)	11 (45)	11 (45)	10 (40)	10 (40)
*Amikacin*	22 (73)	22 (73)	24 (75)	20 (70)	20 (70)	20 (70)	20 (70)	--	--	--
*Streptomycin*	24 (75)	26 (77)	24 (75)	21 (71)	25 (76)	21 (71)	21 (71)	--	--	--
*Ketoconazole*	--	--	--	--	--	--	--	16 (63)	18 (67)	18 (67)
*Amphotericin B*	--	--	--	--	--	--	--	15 (60)	17 (65)	17 (65)

A → (-) Escherichia coli, B → (-) Salmonella enteric serovar typhi, C → (-) Chromo bacterium violaceum, D → (-) Pseudomonas aeruginosa, E → (-) Shigella flexneri, F → (+) Staphylococcus Aureus, G → (+) Bacillus cereus, H → Aspergillus niger, I → Candida albicans, J → Mucor indicus. Standard drugs for Bacterial strains: Amikacin and Streptomycin Standard drugs for fungal strains: Ketoconazole and Amphotericin B. [Control (DMSO) = 6 mm].

**Table 9 biomolecules-12-01883-t009:** The cytotoxicity of ligand (**HL**), mixed ligand complexes (**1b**–**2b**) and standard drug (**cisplatin**) against cancer and normal cell lines.

Compounds	IC_50_ (µM)			
	A549	HepG2	MCF-7	NHDF
**Cisplatin**	31.9 ± 1.6	22.9 ± 1.1	20.2 ± 1.0	26.9 ± 1.3
(**HL**)	126.4 ± 6.3	108.4 ± 5.4	105.2 ± 5.3	208.6 ± 10.4
(**1b**)	29.7 ± 1.2	30.9 ± 1.2	31.7 ± 1.3	72.6 ± 2.9
(**2b**)	32.1 ± 1.3	33.4 ± 1.4	32.5 ± 1.3	74.5 ± 3.0

Average IC_50_ values from at least three independent experiments for drug concentration µM of 50% cell death following 72 h exposure. A549 → Human lung cancer cell line, HepG2 → Liver cancer cell line, MCF-7 → Breast cancer cell line and NHDF → Normal human dermal fibroblasts cell line. Error limits ± 2.5–5.0% (*p* ≤ 0.05).

## Data Availability

Publicly available datasets were analysed in this study. This data can be found here: https://doi.org/10.1039/c8ra09218d (accessed on 9 November 2022) and https://doi.org/10.1016/j.jinorgbio.2022.111953 (accessed on 9 November 2022).

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
