# Peer review of "An Integrated Analysis of Mechanistic Insights into Biomolecular Interactions and Molecular Dynamics of Bio-Inspired Cu(II) and Zn(II) Complexes towards DNA/BSA/SARS-CoV-2 3CLpro by Molecular Docking-Based Virtual Screening and FRET Detection"

_biomolecules, 2022, doi:10.3390/biom12121883_

Round 1
Reviewer 1 Report (New Reviewer)
In this manuscript, the authors have developed bioactive metal complexes and studied their binding with DNA and BSA using several techniques such as absorption and fluorescence spectroscopy, circular dichroism spectroscopy and electrochemical titration. Chemical reactivity of these complexes has been supported by DFT and molecular docking study. Furthermore, they have shown antimicrobial properties and cytotoxic effects of these compounds and compared them with other chemical compounds. They have found that their compound has better antimicrobial activity that the other existing tested compounds. Therefore, this study could be interesting in the field.
The study has been done in detail and the conclusions of the paper have been clearly supported by the results. However, I don’t find this study to be novel enough for publishing in this journal at this stage. Therefore, I would recommend accepting the manuscript after major revisions.
Major points
11. In the antimicrobial activity study, the ligand itself show similar activity like the complex 1b and 2b and the ligands seems to be quite well known. Hence, the authors should study in presence of few more commonly used antimicrobial agents such as penicillin, doxycycline, and tetracycline, and compare their result to establish the usefulness of their developed compounds.
Minor comments
11. Motivation of the study should be clear in the initial few sentences of the abstract.
22. The goal for studying interaction between the complexes, what the authors have developed, and BSA is not clear. This point should be clarified.
33. In the oxidative cleavage experiment in presence of H2O2, what percentage of H2O2 has been used?
44. In Fig. 2, for lane 3, how does the author conclude that complete cleavage has occurred?
55. In table 2, the unit of entropy should be corrected.
66. In page-15, EB has been stated, but its full name mentioned afterwards. State the full form when it is written for the first time.
77. In Fig. 5, the figure legends need to be more clarified. Mention the concentrations of different colored traces and excitation wavelength.
88. In section 3.2.6., Foster theory-based FRET computation is written in detail. Since this is known for long time, it may not be required. Describing the results could be fine.
99. In Fig 6, the legend is not clear. The blue traces are very confusing. It has been stated as emission spectrum and in the figure emission wavelength has been mentioned to be 350 nm. Clarify this.
110. In S21, mention the concentration of the quencher in the figure.
111. In the SI, the fitting parameters have been mentioned up to 4-5 decimal places. This is irrelevant. Consider the error of estimation to choose the decimal place.
112. In Fig. S23 and S27, the solid line is misleading because it is non-linear but in the figures slope and intercept have been mentioned. Clarify this.
113. In Fig. S25, mention the concentration of BSA and emission wavelength in the figure legend.
Author Response
30th November 2022
Journal: Biomolecules Journal (ISSN: 2218-273X)
Publication : MDPI publications
Manuscript ID: Biomolecules-2057922
Title:
An integrated analysis of mechanistic insights into biomolecular interactions and molecular dynamics of bio-inspired Cu(II) and Zn(II) complexes towards DNA/BSA/SARS-CoV-2 3CLpro by molecular docking-based virtual screening and FRET detection
Reply to the Reviewer: 1
please find the attachment

Reviewer 2 Report (New Reviewer)
With the changes that the authors have introduced in the manuscript, this paper can be accepted for publication in Biomolecules, but is necessary a minor revision:
1) Line 302, change "executed" for "recorded". Revise all manuscript
2) Line 321, add formules: hidroxyl (OH-), super oxide (O2(-)), and nitric oxide (N2O)
3) Line 403, change "generation" for "formation". Revise all manuscript
4) Line 523, change "utilized" for "allows"
5) Rewrite: Part 3.2.4
6) Revise Line 824
7) Line 1138-1139, Revise the paragraph "The titration....(Fig. 7 & Table 7)
8) Line 1303, delete "examined"
9) Line 1308, delete "help"
10) Part 3.7, is Cu+ or Cu2+? Because the comparation between Cu+ and Zn2+?
11) Line 1486, change (1b-2b) for (1b and 2b)
12) Line 1491, change "assin" for "indicate"
13) Line 1500, change "demonstrated" for "suggest"
Author Response
30th November 2022
Journal: Biomolecules Journal (ISSN: 2218-273X)
Publication : MDPI publications
Manuscript ID: Biomolecules-2057922
Title:
An integrated analysis of mechanistic insights into biomolecular interactions and molecular dynamics of bio-inspired Cu(II) and Zn(II) complexes towards DNA/BSA/SARS-CoV-2 3CLpro by molecular docking-based virtual screening and FRET detection
Reply to the Reviewer: 2
please find the attachment

Reviewer 3 Report (New Reviewer)
Comments and Suggestions for Authors:
“The manuscript (Biomolecules-2057922) entitled “An integrated analysis of mechanistic insights into biomolecular interactions and molecular dynamics of bio-inspired Cu(II) and Zn(II) complexes towards DNA/BSA/SARS-CoV-2 3CLpro by molecular docking-based virtual screening and FRET detection” written by Karunganathan Sakthikumar et al., In this manuscript, the synthesis of 2-(4-morpholinobenzylideneamino)phenol) ligand and its mixed-ligand complexes (1b) [CuII(L)2(phen)] and (2b) [ZnII(L)2(phen)] and they have well been structurally analyzed by UV-Vis, FT-IR, TGA, EPR, 1H NMR, 13C NMR, ESI-Mass, X-ray diffraction for single crystal, magnetic measurements etc., Bimolecular interaction studies for DNA and BSA were also well studied by electronic absorption spectra, fluorescence spectra, viscometry, cyclic voltammetry, and gel electrophoresis techniques including computational investigation properties. It is accepted that the copper complex (1a) had the best biological efficacy in all cases, including in vitro cytotoxic properties. Moreover, the manuscript is well written, with scientific reasons and suitable literature. After overall peer review, concluded that it is worth publishing in Biomolecules Journal (MDPI)) after a minor reversion.
1. The manuscript is well designed and well written, with the adequate experimental and theoretical results that are supported by adequate literature. DNA/BSA binding properties of all compounds are investigated by electronic absorption spectra, fluorescence spectra, cyclic voltammetry, and gel electrophoresis techniques and further supported by theoretical analysis. It could be useful to perform viscosity experiments too.
2. The author also encouraged to mention the possible mechanism of anticancer properties for test compounds. But, it is not compulsory.
3. What is reason for only selection of BSA protein? Why not HSA?. Explain.
4. Are you sure that the toxic effects are due to the complex? It is possible that complex undergoes hydrolysis reaction and toxicity is due to excess of copper ions? Did you verify cytotoxicity against lung, breast and cervical normal cells as control beside cancer line? With this limitation, the results are worth to be published and the manuscript could be accepted for publication, after a minor revision.
Author Response
30th November 2022
Journal: Biomolecules Journal (ISSN: 2218-273X)
Publication : MDPI publications
Manuscript ID: Biomolecules-2057922
Title:
An integrated analysis of mechanistic insights into biomolecular interactions and molecular dynamics of bio-inspired Cu(II) and Zn(II) complexes towards DNA/BSA/SARS-CoV-2 3CLpro by molecular docking-based virtual screening and FRET detection
Reply to the Reviewer: 3
please find the attachment

Round 2
Reviewer 1 Report (New Reviewer)
The authors have addressed all the comments very well. I will recommend to accept the manuscript as it is.
This manuscript is a resubmission of an earlier submission. The following is a list of the peer review reports and author responses from that submission.
Round 1
Reviewer 1 Report
The manuscript entitled "An integrated analysis of mechanistic insights into biomolecular interactions and molecular dynamics of bio-inspired Cu(II) and Zn(II) complexes towards DNA/BSA/SARS-CoV-2 3CLpro by molecular docking-based virtual screening and FRET detection" written by Sakthikumar et al. describes the synthesis of ligand 2-(4-morpholinobenzylideneamino)phenol as well as Cu(II) and Zn(II) mixed-ligand complexes 1b and 2b. Ligands used for metal complexation are 2-(4-morpholinobenzylideneamino)phenol) and 1,10-phenanthroline. The work is the appendix of the previous research (Sakthikumar et all, J. Inorg. Biochem. 2022, 236, (111953) 1–23). Prepared ligand and complexes are characterized by UV-Vis spectroscopy, determination of magnetic moments, thermo gravimetric analysis and X-ray crystallography. Complexes as well as ligand bind to DNA and BSA, where the Cu(II) complex showed the highest affinity toward biomolecules and efficient DNA cleavage. Experimental findings were supported by molecular docking. Finally, among examined compounds, Cu(II) complex showed the best antimicrobial and antioxidant properties, as well as cytotoxic properties.
There is a lot of both experimental and computational work reported in this manuscript, yet I do have concerns with the following points:
1. Manuscript is too long and consequently difficult to read; for example, 3.2.1. Analysis of DNA cleavage: only discussion of own results should be retained. The description and explanation of the methods at beginning of every paragraph are far too extensive, so results and discussion get lost in the theoretical details; Fig 3 is unnecessary, all equations are also not necessary in the main text, and they could be moved to ESI. At the other hand also ESI contains a discussion on the NMR spectra, theoretical explanation of DTm method, FRET, etc., which is unnecessary. The manuscript should be shortened and rewritten more concisely.
2. Even though extensive experimental work being done, some important details are missing or incorrect, for example:
· Since the important goal of this paper is an investigation of metal complexes and their interaction with biomolecules, the characterisation of studied metal complexes in a biologically relevant aqueous buffer is necessary. Although 2b doesn’t absorb visible light, its UV spectra should be attached, as well as UV Vis spectra of HL and 1b in water media, besides their spectra in methanol. Also, concentration dependence of absorbance should be given.
· Spectra of 1b (Figure S9) is unacceptable and should be measured again, using lower concertation and/or shorter optical path length.
· Relatively high concentrations of compounds were used in experiments (for example, 50 μM in UV-Vis titrations), and authors should provide evidence that their compounds (having large aromatic moieties) don’t aggregate. There is no evidence that they have monomer species in water solution. The possibility of dimerization, and aggregation, has to be considered.
· For calculations of binding constants, the authors use É›f → extinction coefficient of free compound (Table 1), but they didn’t give the values or claimed how they got the value. This value cannot be calculated from the single spectra but from the linear dependence of the absorbance vs concentration. There is no evidence that they have monomer species in water solution.
· There are many problems considering thermal melting curves (Fig.S18.):
Tm values in the text have the unit °C; while in the table Kelvins were used, which is inconsistent, but this is a minor point.
There is no plateau at the beginning of the experiment, the concentration of DNA was not noted, and it looks like there is more than one transition even for ct-DNA (problems with precipitation and/or purity, variety in polynucleotide chain length?), the line-symbol presentation looks very odd (are absorbancies recorded every five degrees?), it is not clear how melting temperatures were determined (first derivation? tangent method?).
Finally, DTm values in Table 2 are not correct: for example, 343.0 K - 337 K (ctDNA) is not 279.0 K!
For more useful information about thermal melting, I recommend the reference: J.L. Mergny, L. Lacroix, Analysis of thermal melting curves, Oligonucleotides, 13 (2003) 515-537.
· 3.2.2. Assessment of DNA binding properties using UV–Vis absorption titration: Spectra of HL (titration, Figure 4) doesn’t match to HL spectra on Fig. S9 (here is no broad band at ca 440 nm)
· It is not clear which excitation wavelength lexc was used for EB in the displacement experiment. Do 1b and 2b absorb at this lexc? Also, it is important to check if EB binds to any of the compounds (HL, 1b or 2b).
· In Table 4 title there are BSA constants mentioned, but there are no BSA constants in the table. Although authors claim that compounds have low absorbance at excitation wavelength 278 nm (BSA fluo-titration), it is clear that at least 1b absorbed UV light at this wavelength (Figure S9). Again, it is crucial to give absorbance–concentration dependence. So fluorescence quenching upon addition of compound was not caused only by complexation, but also by an inner filter effect. Equation 12 was not from reference 75; anyhow, if employed, concrete values used in this equation should be given. (Table 4, Figure 5)
· Figure4: Titration 1b and 2b with ct-DNA: There is something wrong in titration experiment or presentation since DNA has strong absorbance at 260 nm. There has to be an increase of absorption in the range 250-300 nm with the addition of an excess of DNA, surely not hypochromism.
Obtained n values are too high for intercalation, taking into account neighbor-exclusion principle
3. ESI 3.1.1. Synthesis of Schiff base ligand (HL): besides amount (mol) masses of compounds (mg) should be given. All NMR signals (1H and 13C) and IR bands and their assignation should be given separately for every particular compound. Comparison and discussion can be moved to the main text, if necessary.
4. Some references are missing, for example:
This statement (ΔGb° values as evidence for intercalation) should be supported with reference: „Moreover, the overall observed ΔGb° values in all cases were in the range of from –22.50 to –25.12 kJmol–1 (Table 1), which also indicates that the compounds spontaneously intercalate to DNA.“
„The result reveals the electrostatic or groove binding mode when ΔTm < 8 °C“. I’m not sure if it is correct, reference is needed.
Consequently, I would not recommend publication in its present stage unless major modifications are made.
Author Response
Journal: Biomolecules Journal (ISSN: 2218-273X)
Publication : MDPI publications
Manuscript ID: Biomolecules-1938741
Title:
An integrated analysis of mechanistic insights into biomolecular interactions and molecular dynamics of bio-inspired Cu(II) and Zn(II) complexes towards DNA/BSA/SARS-CoV-2 3CLpro by molecular docking-based virtual screening and FRET detection
Reply to the Reviewer:1
please find the attachment

Reviewer 2 Report
The article by Rui Werner Maçedo Krause et al, entitled ‘ An integrated analysis of mechanistic insights into biomolecular interactions and molecular dynamics of bio-inspired Cu(II)and Zn(II) complexes towards DNA/BSA/SARS-CoV-2 3CLpro by molecular docking-based virtual screening and FRET detection’ suggests bioinspired CU(II) and Zn(II) possess antimicrobial and cytotoxic properties , further suggested that coordination increased the activity of these complexes .
Overall the work sounds technically good , there are no major points for this work, with minor corrections and justification for the following queries, this can be published.
1. Authors haven’t explained the significance of binding studies with BSA? why authors choose BSA instead of HSA ??? since HSA is the protein that exists in the humans! not BSA? please justify ?
2. Authors failed to explain the correlation between the experiments in terms of overall outcome of the work presented in the manuscript, for example, why DNA and BSA binding studies are relevant in terms of antimicrobial and cytotoxic properties???
3. Figures are of low quality and should be improved in terms of resolution.
4. English can be improved by avoiding lengthy statements which might be difficult to comprehend for the reader
Author Response
31st October 2022
Journal: Biomolecules Journal (ISSN: 2218-273X)
Publication : MDPI publications
Manuscript ID: Biomolecules-1938741
Title:
An integrated analysis of mechanistic insights into biomolecular interactions and molecular dynamics of bio-inspired Cu(II) and Zn(II) complexes towards DNA/BSA/SARS-CoV-2 3CLpro by molecular docking-based virtual screening and FRET detection
Reply to the Reviewer:2
please find the attachment

Round 2
Reviewer 1 Report
I find manuscript unacceptable in its present stage.
Manuscript was shorten insignificantly, there is still far too much literature data (not only specific, but textbook data) besides introduction, which is irrelevant for discussion of their own results.
I am not sure if authors did actually understand some queries,
For example, it was not „concentration dependence titration“ (DNA titrated with compound solution) what was required, but simple UV-Vis spectra of compounds (HL, 1b, 2b) in buffer.
They didn't provide electronic spectra (UV-Vis ) of compounds in buffer, and also there is no spectra of 2b in methanol (it is clear that 2b doesn't absorb in visible region, but it should have UV spectra considering ligands).
Cover letter 1, Figure 2: It was not clear which wavelength was presented, also molar extinction coefficients should be presented (together with the units and wavelengths) since they are used for calculation of stability constants.
DTm values are wrong again: D Tm is difference between melting temperature of complex and DNA alone: this temperature difference is equal, no matter if represented in Kelvin or Celsius degrees (Table 2). (D Tm values in Kelvin are calculated wrongly again)
Values represented in Table 2 don't match to those in Figure 4.
Tm values (Table 2) for 1b and 2b are the same as in the first version of manuscript, although there was new experiment performed.
EB Displacement experiment, excitation:
Ans:-„Also, in the absence and presence of rising quantities of each test compound, the fluorescence emission spectra of the EB-DNA adduct was examined at 610 nm“….
Besides long answer, there is still no answer about excitation wavelength of EB that was used for displacement experiment, and no answer if compounds HL, 1b, 2b absorb at that wavelength.
Further, I really don't understand why authors add this paragraph to their answer:
Ans:-:Safety Measures: Ethidium bromide is a mutagen and strong intercalator with DNA and RNA and after intercalation, which enhances fluorescence effect. Meanwhile, EB is probable carcinogen and toxic. Hence, we should handle it with much more attentions and should be stored in a cool, dark place away from strong oxidizing agents. Always keep the container tightly closed when not in use. Eyes: If contact with eyes, immediately flush with copious amounts of cool water for at least 15 minutes. Remove contacts lenses if possible.
Titration of compounds with DNA, hypochromic effect in the range 250-300 nm:
Q.2 (ix). Figure 4: Titration 1b and 2b with ct-DNA: There is something wrong in titration experiment or presentation since DNA has strong absorbance at 260 nm. There has to be an increase of absorption in the range 250-300 nm with the addition of an excess of DNA, surely not hypochromism. Obtained n values are too high for intercalation, taking into account neighbor-exclusion principle.
Ans:- Thank you for your comment and appreciation. It is correct that DNA has strong absorbance at 260 nm, the wavelength can also monitor while adding the test compounds in the DNA solution. This is experiment is DNA based test compound addition titration. But, our experiment is test compounds based DNA addition titration.
This is the exact reason why absorbance had to be increased in the range 250-300 nm, regardless intercalation, due to huge excess of DNA over compound. Hypochromic effect as result of intercalation could be expected at the wavelengths longer than 300 nm. Therefore I think there is something wrong with titrations represented in Figure 3, especially since there is no representation of absorbance change at single wavelength.
Considering n values:
Ans:-We have also corrected the n values (number of binding site) in table 4 (Main text page no: 28).
The only value that was changed in Table 4 (compared to first version of the text) is for the complex of HL with DNA, calculated by Scartchard and it is again too high for intercalation. However, even bigger problem is that n value is changed, while stability constant is the same as in the first version. This is impossible if the values are actually calculated, because ratio (n=[bound compound] / [polynucleotide]) and stability constant (Ks) are mutually dependent: change of n will surely reflect the constant.